# Uncertainties and discrepancies in the representation of recent storm surges in a non-tidal semi-enclosed basin: a hind-cast ensemble for the Baltic Sea

Marvin Lorenz[1] and Ulf Gräwe[1]

[1]Leibniz Institute for Baltic Sea Research Warnemünde, Rostock, Germany

**Correspondence:** Marvin Lorenz (marvin.lorenz@io-warnemuende.de)

**Abstract.** Extreme sea level events, such as storm surges, pose a threat to coastlines around the globe. Many tide gauges have been measuring sea level and recording these extreme events for decades, some for over a century. The data from these gauges often serve as the basis for evaluating the extreme sea level statistics, which are used to extrapolate sea levels that serve as design values for coastal protection. Hydrodynamic models often have difficulty in correctly reproducing extreme sea levels and, consequently, extreme sea level statistics and trends. In this study, we generate a 13-member hind-cast ensemble for the non-tidal Baltic Sea from 1979 to 2018 using the coastal ocean model GETM (General Estuarine Transport Model). In order to cope with mean biases in maximum water levels in the simulations, we include both simulations with and without wind speed adjustments in the ensemble. We evaluate the uncertainties in the extreme value statistics and recent trends of annual maximum sea levels. Although the ensemble mean shows good agreement with observations regarding return levels and trends, we still find large variability and uncertainty within the ensemble (95% confidence levels up to 60 cm for the 30-year return level). We argue that biases and uncertainties in the atmospheric reanalyses, e.g. variability in the representation of storms, translate directly into uncertainty within the ensemble. The translation of the variability of the 99th percentile wind speeds into the sea level elevation is in the order of the variability of the ensemble spread of the modelled maximum sea levels. Our results emphasise that 13 members are insufficient and that regionally large ensembles should be created to minimise uncertainties. This should improve the ability of the models to correctly reproduce the underlying extreme value statistics and thus provide robust estimates of climate change-induced changes in the future.

## 1   Introduction

Extreme sea levels (ESLs) result from complex interactions between different processes that lead to very high sea levels that can cause coastal flooding, often damaging coastal communities. It is therefore important to understand and assess how sea-level dynamics and ESLs have changed in the past and how they will change in the future.

For the non-tidal Baltic Sea, the Baltic Sea Assessment Reports (BEARs, https://esd.copernicus.org/articles/special_issue1088. html) provide an overview of the current state of knowledge and knowledge gaps regarding sea-level dynamics and ESLs in Weisse et al. (2021) and Rutgersson et al. (2022), as well as projected changes of mean sea level and ESLs in the future in Meier et al. (2022a, b). Weisse et al. (2021) argue that improved numerical models with higher resolution are needed since

the occurrence of ESLs is not always well described by classical statistical distributions (Männikus et al., 2019). However, the representation of ESLs in numerical models is depending on the representation of storms in the atmospheric dataset, adding another source of uncertainty. In this study, we will assess the uncertainty and variability that can be expected from numerical models when investigating simulations of ESLs, focusing here on hind-cast simulations.

The most prominent ESLs in the Baltic Sea are storm surges. However, even during storm surges, other effects such as wave
setup and seiches can simultaneously modulate the sea level. For the Baltic Sea, *preconditioning*, i.e., the mean filling of the Baltic Sea, plays an important role. It describes the process of mean sea level changes due to water mass exchange with the North Sea caused by persistent westerlies or easterlies on a time scale of weeks (e.g. Madsen et al., 2015; Soomere and Pindsoo, 2016; Weisse and Weidemann, 2017). This significantly increases or decreases the background sea level upon which the storm surges act and thus the overall height of the ESL event. One of the highest sea levels observed in the Baltic Sea resulted from
the coincidence of preconditioning that filled the Baltic Sea to very high levels and the simultaneous occurrence of a storm surge. This was in 1872 when the maximum water levels in the western Baltic Sea along the German coast exceeded 3 m above the mean sea level (Rosenhagen and Bork, 2009; Feuchter et al., 2013; Hofstede and Hamann, 2022). The return period of this event is estimated to be $\sim 3400$ years (Bork and Müller-Navarra, 2009). Recently it has been shown that this ESL could have been even higher if the preconditioning and the weather systems had been slightly different (Andrée et al., 2023).

Sea-level rise is projected to increase the global mean sea level by 28 to 102 cm (median) by 2100, depending on anthropogenic greenhouse gas emissions as well as the response of the ocean and the cryosphere in the upcoming decades (IPCC 2021, Chapter 9, Fox-Kemper et al., 2021). Considering ice sheet dynamics in more detail, Bamber et al. (2019) list in their Tab. 2 possible ranges from 69 to 111 cm (median), 49-98 to 79-174 cm (17-83%), and 36-126 to 62-238 cm (5-95%). The sea-level rise is expected to be additive to the ESLs in the open Baltic Sea (Gräwe and Burchard, 2012; Hieronymus et al.,
2018). Recent scenario studies for the Baltic Sea have shown no significant changes in maximum wind speeds (Christensen et al., 2022) from which Meier et al. (2022a) conclude that ESLs relative to mean sea level do not change beyond statistical significance.

Past mean sea-level rise for the Baltic Sea has been shown to align with the global mean (Weisse et al., 2021, and references therein). However, sea-level rise has been spatially heterogeneous due to the glacial isostatic adjustment (GIA, Peltier, 2004)
and will continue to be heterogeneous in the future (Grinsted, 2015). In some areas, the GIA exceeds the sea-level rise, e.g. in the Gulf of Bothnia, leading to decreased water depth. Mean sea-level rise in the Baltic Sea also depends on atmospheric changes, such as atmospheric pressure, wind speed, and wind direction (Gräwe et al., 2019). In general, sea-level rise shifts the ESLs (relative to present-day mean sea level) to higher base levels, and thus high ESLs become more frequent, although surge events themselves may not have changed their frequencies (Wahl et al., 2017). Present sea-level rise and atmospheric
changes have already led to more frequent and longer ESL events (Wolski and Wiśniewski, 2021). Furthermore, ESLs have also become significantly higher in the eastern Baltic Sea, e.g., up to 5 to 10 mm yr$^{-1}$ along the Estonian coast and the Gulf of Finland (Pindsoo and Soomere, 2020). Because of these changes, non-stationary statistical tools have been used to infer further these trends, e.g. non-stationary distributions of the Generalised Extreme Value (GEV) (Coles et al., 2001). For the Gulf of Riga, Kudryavtseva et al. (2021) show that the shape parameter has changed significantly and often abruptly in the past. A

sudden decrease of the shape parameter between 1986 and 1990 may be related to the North Atlantic Oscillation (NAO). High correlation values indicate a direct connection between the NAO and the mean sea-level of the Baltic Sea (e.g. Andersson (2002); Hünicke and Zorita (2006); Karabil et al. (2018)), winds (e.g. Lehmann et al. (2011)), and hence ESLs (e.g. Suursaar and Sooäär, 2007; Wolski and Wiśniewski, 2023). However, extreme winds show little correlation with the NAO (Bierstedt et al., 2015).

Due to all these different components leading to the observed ESLs, not only do numerical models have difficulty correctly reproducing each ESL in hind-cast simulations, but these uncertainties and biases are also present in scenario simulations for the future. Since most atmospheric processes cause sea level changes, high-resolution (both in time and space) atmospheric forcing datasets are required to reproduce the observed ESLs. In modern atmospheric reanalyses, data assimilation is used to force the model to reproduce observed weather systems such as passing storms. Nevertheless, data assimilation does not make the atmospheric datasets perfect. Depending on how the models are configured and the focus of the respective simulation, each reanalysis has its strengths and weaknesses. Global reanalyses have a coarse resolution, and regional reanalyses depend on the boundary conditions of the global reanalyses. To cope with these issues, ensembles are created from as many members as possible to compensate for the strengths and weaknesses of each member by examining the ensemble mean and the ensemble variability. However, ESL hind-cast simulations are sparse, and hind-cast ensembles focusing on ELSs are basically non-existent or include only a few members. Global ESL reanalyses (e.g. Muis et al., 2016) have the disadvantage that they cannot focus and adjust for local biases. Regional hind-casts can be adjusted and benefit from regionally down-scaled atmospheric datasets. However, these datasets are rare. Alternatively, data-driven models can be used to generate an ensemble (e.g. Tadesse and Wahl, 2021; Tadesse et al., 2022). However, these data-driven models are only applicable to gauge stations. Numerical models are needed to fill the spatial gaps and to untangle underlying physical processes.

In this study, we generate a hind-cast ensemble for the Baltic Sea to investigate the ability of the ensemble to reproduce observed ESLs and trends. We will focus solely on the variability of the representation of Baltic Sea ESLs within the ensemble and trends in annual maximum sea levels. We will examine only ESLs caused by atmospheric processes. We use different extreme value distribution methods, the Generalised Extreme Value Distribution (GEV) and the Generalised Pareto Distribution (GPD), to evaluate the variability of expected return levels. We show that the variability within state-of-the-art atmospheric hind-cast products is large for extremes. Therefore, reproducing observed ESLs, estimating return levels, and recent trends is challenging for numerical simulations. We use six atmospheric reanalyses to compile an ensemble of ESL simulations in the Baltic Sea from 1979 to 2018.

## 2 Methods

### 2.1 Numerical Simulations

We use the General Estuarine Transport Model (GETM (version 2.5), Burchard and Bolding, 2002), a structured coastal ocean model (Klingbeil et al., 2018), to simulate the surface elevation in the Baltic Sea. The model has been widely used for many Baltic Sea applications, including storm surges: Gräwe and Burchard (2012). Here, we use GETM to solve the

vertically integrated shallow water equations, considering only barotropic effects and ignoring baroclinic processes (except for one simulation where we include static baroclinic circulation). We employ the superbee advection scheme (Pietrzak, 1998). The wind stress is calculated from the 10 m wind fields with

$$\boldsymbol{\tau} = C_D \rho_{\text{air}} |\boldsymbol{u}_{10}| \boldsymbol{u}_{10}, \tag{1}$$

with $\rho_{\text{air}} = 1.25 \text{ kg m}^{-3}$ and the 10 m wind speed vector $\boldsymbol{u}_{10}$. The drag coefficient $C_D$ is computed by the formulation of Kara et al. (2000). In our barotropic simulations we do not consider temperature differences between the ocean and atmosphere. Therefore, the drag coefficient reads as

$$C_D = 10^{-3} \left( 0.862 + 0.088w - 0.00089w^2 \right),$$

where the wind speed is limited: $w = \max(2.5, \min(32.5, |\boldsymbol{u}_{10}|))$. The numerical setup of the Baltic Sea has a resolution of one nautical mile, which is nested into a coarse (5 n.m.) model of the Northwest Atlantic with a bottom roughness of $z_0 = 0.005 \text{ m}$, see Fig. 1, similar to the setups of e.g. Gräwe et al. (2015). The Northwest Atlantic model is forced with ERA5 (Hersbach et al., 2020) with unchanged wind speeds. Along its boundary, air pressure-induced water level changes are imposed using the atmospheric pressure from ERA5 to include large pressure systems from the Atlantic into the model chain (inverted barometric effect). This simulation prescribes boundary conditions for the one nautical mile North Sea / Baltic Sea domain (all simulations). This may introduce some inconsistencies in the sea level when using other atmospheric forcings for the North Sea / Baltic Sea nesting stage. However, we assume these errors to have little effect on Baltic Sea ESLs since the Danish Straits act as a low-pass filter. Thus, ESLs are generated inside the Baltic Sea as a superposition of storm surges, filling states, and local seiches which are all included in the model setup. Note that the boundary conditions for this nesting do not include tides to avoid introducing tide-surge interactions (Arns et al., 2020). Tides are negligible for the Baltic Sea, as explicitly shown by Gräwe and Burchard (2012). However, standing-wave-surge (seiche-surge) interactions (Arns et al., 2020) are included. Furthermore, the mean sea level is kept constant in order to study only the atmospheric-induced extreme sea levels themselves (ESL distributions treat the mean sea level as a linear term Coles et al. (2001)). The North Sea / Baltic Sea domain is set up with a bottom roughness of $z_0 = 0.001 \text{ m}$, a value that is also used by other coastal ocean models than GETM for the Baltic Sea (e.g. Kärnä et al., 2021).

High-resolution atmospheric forcing in both time and space is required to model extreme sea levels. For the Baltic Sea, we consider the regional atmospheric reanalyses listed in Tab. 1, which provide the needed high-resolution fields for winds speeds (10 m height) and sea level pressure. For this study, we investigate the period from 1979 to 2018, the overlap between all atmospheric datasets, focusing on the variability between these ensemble members. We included one run ('UERRA baroclinic') where we included monthly mean static density fields and static ice cover (taken from Gräwe et al. (2019)). For each atmospheric forcing, we ran one simulation with default wind speeds (directly taking the values provided by the atmospheric dataset) and one with increased wind speeds, see Tab. 1, for a total of 13 ensemble members. Before all analyses, we subtracted the linear trends from each time series to explicitly consider only ESLs relative to the respective mean sea level. This excludes changes in the mean sea level due to persistent changes in wind speed, wind direction, sea level pressure changes, and external loading (Gräwe et al., 2019).

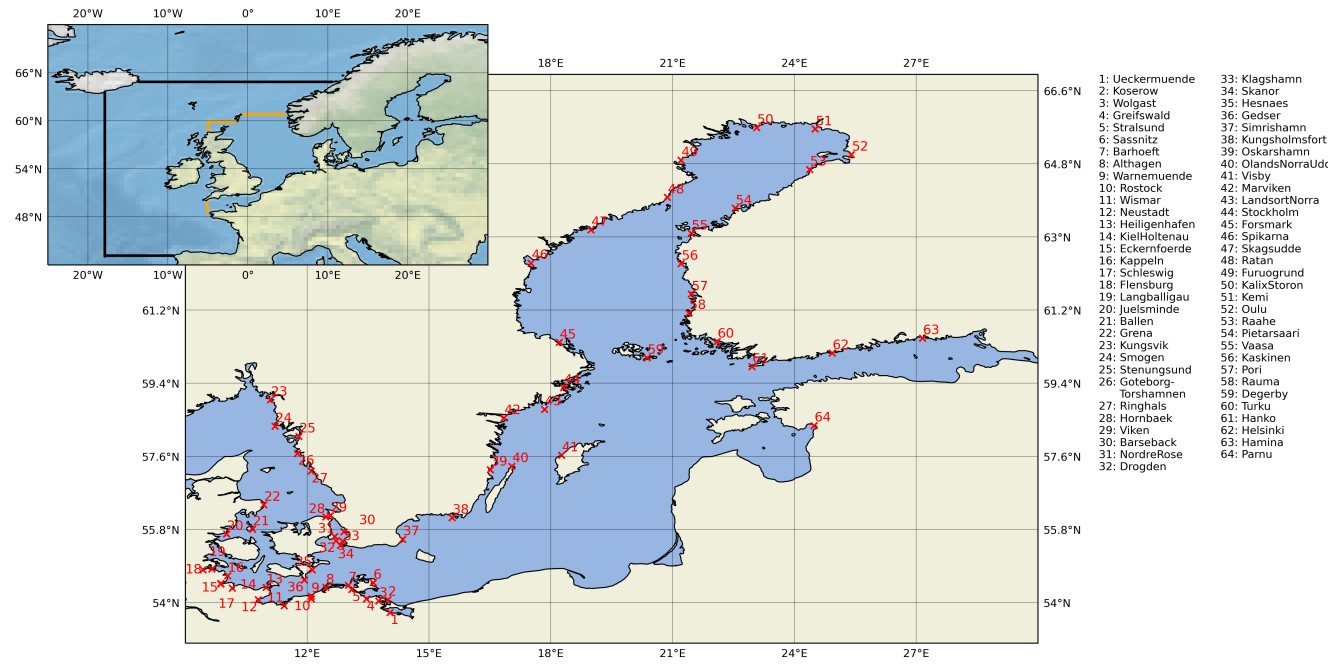

**Figure 1.** Model domain and locations of the stations (red crosses and numbers) used for validation. The black line indicates the boundary of the coarse Northwest Atlantic Ocean domain. The orange lines mark the boundaries of the one nautical mile setup of the North Sea & Baltic Sea.

**Table 1.** Overview of the different atmospheric datasets used in this study. Each atmospheric forcing is included one time with default wind speeds and one time with increased wind speeds except the UERRA (baroclinic) run which is only computed with increased wind speeds.

| Atmospheric forcing / run | time period | temporal resolution | spatial resolution | wind factor increase | reference |
|---|---|---|---|---|---|
| UERRA | 1961 - 2018 | hourly | 11 km | + 7 % | Ridal et al. (2017) |
| UERRA (baroclinic) | 1961 - 2018 | hourly | 11 km | + 7 % | Ridal et al. (2017) |
| ERA5 | 1950 - 2018 | hourly | 32 km | + 11 % | Hersbach et al. (2020) |
| CFSR | 1979 - 2018 | hourly | 32 km | + 3 % | Saha et al. (2010, 2014) |
| coastDat1 | 1950 - 2018 | hourly | 0.22° | + 7 % | Geyer (2014) |
| coastDat2 | 1979 - 2018 | hourly | 11 km | + 4 % | Geyer (2014) |
| coastDat3 | 1948 - 2020 | hourly | 11 km | + 3 % | Geyer et al. (2021) |

## 2.2 Observational Gauge Data

We use several gauges to validate the model results along the Baltic Sea coastline. The data is obtained from the European Marine Observation and Data Network (EMODnet, https://emodnet.ec.europa.eu) and the Global Extreme Sea Level Analysis (GESLA, Woodworth et al., 2016; Haigh et al., 2022). From both sources we have obtained quality controlled sea level data of hourly frequency. The sea level data is accurate to within one centimetre. The station record lengths and data gaps are summarised in Fig. 2 (a table of the gauges is provided in Tab. S1 in the Supplement). Their locations are shown in Fig. 1 and also listed in Tab. S1 in the Supplement. Note that we only included stations with time series starting in 2004 and earlier, and that we use the data from 1979 onwards to match the simulation periods. For each gauge, the mean sea level rise is subtracted (linear fit over the entire time series) before analysis and comparison with model results. This also excludes sea level changes due to GIA from the tide gauges. For the direct comparison of ESL events with observations (Section 3.1), we use the German Federal Maritime and Hydrographic Agency's definition of a storm surge in the Baltic Sea. It defines a storm surge as a sea level of more than 1 metre above mean sea level.

## 2.3 Extreme Value Statistics

We emphasise that we assume that the statistical properties of the extreme value distributions are constant in time within the considered time frame. Therefore, we do not consider temporal changes in the extreme value distributions. This assumption may not be valid (Kudryavtseva et al., 2018, 2021). However, this study focuses on the variability within the ensemble and not the comparison between static and temporally varying extreme value distributions.

### 2.3.1 Generalised Extreme Value Distribution (GEV)

To describe the distribution of storm surge heights and return periods, we use the general extreme value (GEV) distribution (Coles et al., 2001) using the time series of annual storm season (July to June) block maxima (Männikus et al., 2020). The GEV (cumulative distribution function) is defined by

$$F(z, \mu, \sigma, \xi) = \exp\left\{-\left[1 + \xi\left(\frac{z-\mu}{\sigma}\right)\right]^{-1/\xi}\right\}, \tag{2}$$

where $z$ is the sea level, $\mu$ is the location parameter, $\sigma$ is the scale parameter, and $\xi$ is the shape parameter. The shape parameter $\xi$ governs the tail of the distribution and depending on its value the distribution reduces to:

1. Gumbel distribution for $\xi \to 0$:

$$F(z, \mu, \sigma) = \exp\left\{-\exp\left\{-\frac{z-\mu}{\sigma}\right\}\right\}. \tag{3}$$

2. Fréchet distribution for $\xi > 0$:

$$F(z, \mu, \sigma, \xi) = \begin{cases} 0 & \text{for} \quad z \le \mu, \\ \exp\left\{-\left(\frac{z-\mu}{\sigma}\right)^{-\xi}\right\} & \text{for} \quad z > \mu. \end{cases} \tag{4}$$

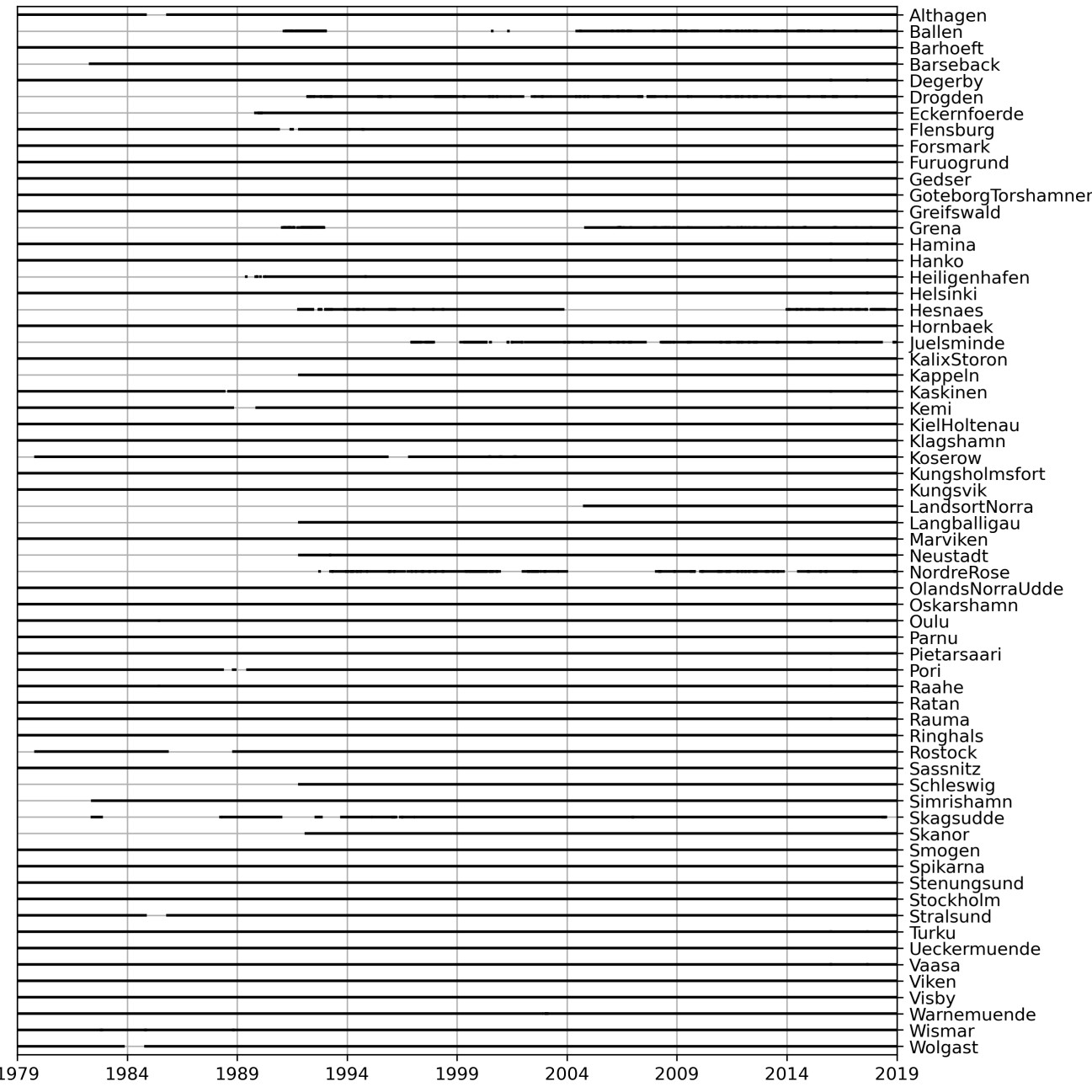

**Figure 2.** Overview of the record lengths and data gaps of the gauge stations. A table with exact locations and more details of the exact time spans and gaps can be found in the Supplement in Tab. 1.

3. Weibull distribution for $\xi < 0$:

$$F(z, \mu, \sigma, \xi) = \begin{cases} \exp\left\{-\left(\frac{z-\mu}{\sigma}\right)^\xi\right\} & \text{for} \quad z < \mu, \\ 1 & \text{for} \quad z \geq \mu. \end{cases} \tag{5}$$

For each gauge used in this study, the GEV is fitted to the annual storm season maxima. We use the Python code from Reinert et al. (2021) for the fitting, which uses the maximum likelihood estimation method.

### 2.3.2 Generalised Pareto Distribution (GPD)

Another function to describe the statistical distribution of extreme values is the Generalised Pareto Distribution (GPD) (cumulative distribution function),

$$F(z, \mu, \sigma, \xi) = \begin{cases} 1 - \left(1 + \frac{\xi z}{\tilde{\sigma}}\right)^{-1/\xi} & \text{for} \quad \xi \neq 0, \\ 1 - \exp\left(-\frac{z}{\tilde{\sigma}}\right) & \text{for} \quad \xi = 0, \end{cases} \tag{6}$$

where $\tilde{\sigma}$ is the scale parameter, $\tilde{\sigma} = \sigma + \xi(u - \mu)$, and $u$ is the threshold (Coles et al., 2001). This method uses a peak-over-threshold to sample the time series (minus the mean sea-level rise). In this study, we use the $99.7\%$ percentile of the sea level distribution as the threshold suggested by Arns et al. (2013) and only consider events separated by more than 48 hours. We use the maximum likelihood estimation method and a python script for the fitting.

## 3 Results

### 3.1 Comparison of sea levels

Since we are interested in the ESLs, we compare the observed ESLs for each station with the modelled ESLs (Fig. 3), where the stations are sorted in a clockwise direction along the Baltic Sea coast, starting from the Western Baltic Sea (Fig. 1). The time series comparison for each tide gauge station shows a good agreement with low Root Mean Square Errors ($\mathrm{RMSE} \leq 0.1\,\mathrm{m}$) and high Pearson correlation coefficients $R \approx 0.9$, see Fig. S1. Furthermore, we only consider events that are separated by more than 48 hours. We search for events that fulfil these criteria in the observed tide gauges and compare them with the modelled sea level for each station. This comparison shows that the default wind speed simulations have a clear negative mean bias, red dots and values in Fig. 3. On average, the simulations underestimate the maximum water levels. The bias is different for each atmospheric forcing. To improve this comparison of ESLs in the model, we increased the wind speeds by a constant factor for each atmospheric dataset, see Tab. 1. With this increase, the mean bias of the model results reduces significantly; see black dots and values in Fig. 3. However, it should be kept in mind that due to the homogeneous wind speed increase, some areas, i.e. stations, in the model domain show a larger bias than before, e.g. the central and Northern Baltic Sea. However, the wind speed increase is necessary to capture the correct ESLs in the Western Baltic Sea. Therefore, in the following analysis of the ESLs, we consider both the default simulations (larger mean bias) and the simulations with increased/adjusted wind speeds (smaller

mean bias) as an ensemble of 13 members. Note that the ESLs for stations in shallow coastal lagoons such as 'Althagen' and 'Schleswig' are overestimated because the model does not resolve the local topography and inlets.

## 3.2 Return levels and variability within the ensemble

### 3.2.1 Return levels obtained from the GEV method

Since the period considered is only 40 years, we compare the 30-year return periods, which should be well estimated by the underlying 40 years of data, see also Gräwe and Burchard (2012). The comparison of modelled and observed 30-year return levels for the stations (Fig. 4) shows a large variability between the atmospheric reanalyses. A more detailed view of all return level fits and the different simulations' confidence intervals is shown in Fig. S2 for the station 'Warnemuende' in the Supplement. We have again sorted the stations clockwise around the Baltic Sea coast, starting in the Western Baltic Sea. This depiction shows that the default wind speed simulations have a clear negative bias in the Western Baltic Sea but show good agreement in the central and Northern Baltic Sea regions. On the other hand, the adjusted wind speed simulations show good agreement in the Western Baltic Sea and an overestimation in the central and Northern regions. This pattern is similar to the general comparison of ESLs in Section 3.1. This shows that the biases from the previous section are directly reflected in the return level estimates. Each atmospheric forcing has its strengths and weaknesses regarding how well storm systems are represented in terms of intensity and tracks in different regions. Therefore, the spatially homogeneous adjustment of wind speeds improves the representation in some areas but deteriorates it in others. However, these problems seem to cancel out for the ensemble (Fig. 4h). For most of the stations, the ensemble mean return level is close to the mean return level based on the observations and well within the respective 95% confidence interval.

Using the model to fill the area between the observed gauges, we show the ensemble mean 30-year GEV return level for every fourth data point (every second point in latitude and longitude) in the Baltic Sea, see Fig. 5a. Since westerlies are the dominant winds in the Baltic Sea region, the eastern coasts show higher return levels as expected and well documented (e.g. Wolski et al., 2014). In the South-Western Baltic Sea, west-facing coasts have higher return levels than those facing north or south but lower expected return levels than the Northern and Eastern Baltic Sea. Although the ensemble mean shows good agreement with the estimated return levels based on the observations, the spread, i.e. the 95% confidence interval, shows large values up to 50 cm and more (Fig. 5b).

Comparing the different ensemble members to the ensemble mean (Fig. 6) we see more clearly how each member has its regions where it is close to the ensemble mean and regions where it deviates strongly. This depiction shows that the general pattern of the high and low return levels is similar in each simulation, but the estimated return levels vary considerably, as already seen previously. Without again describing the spatial pattern of each ensemble member, we want to emphasise the discrepancies between the individual simulations. Generally, the adjusted wind speed simulations are at the high end of the distribution, whereas the default wind speed simulations are at the low end. The simulation with the lowest return levels is ERA5. The simulation with the highest return levels is the UERRA simulation, which includes the increased wind speeds and

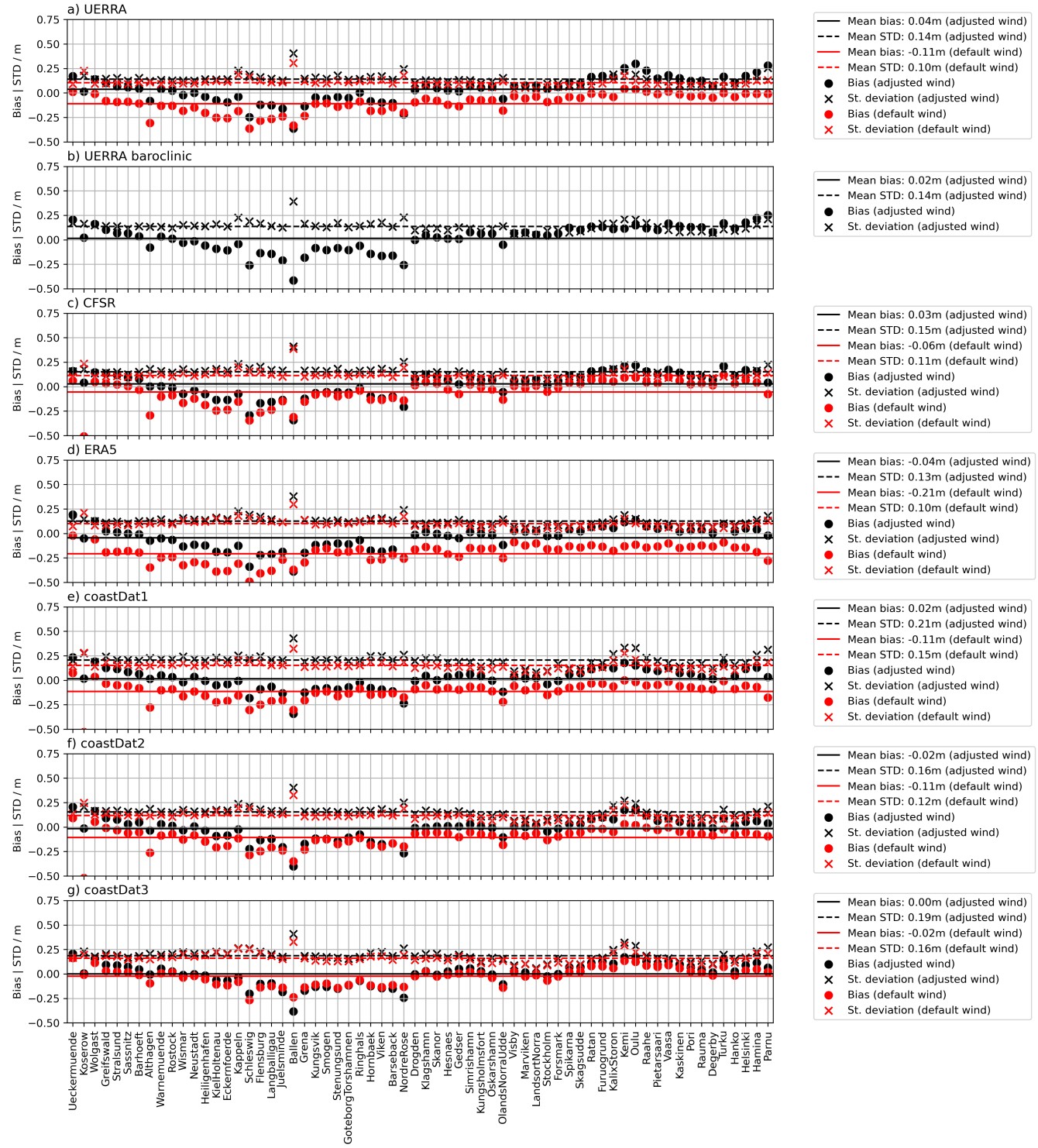

**Figure 3.** Mean bias of the model's ESLs and standard deviation summarised (STD) for each station of Fig. 2 for simulations listed in Tab. 1.

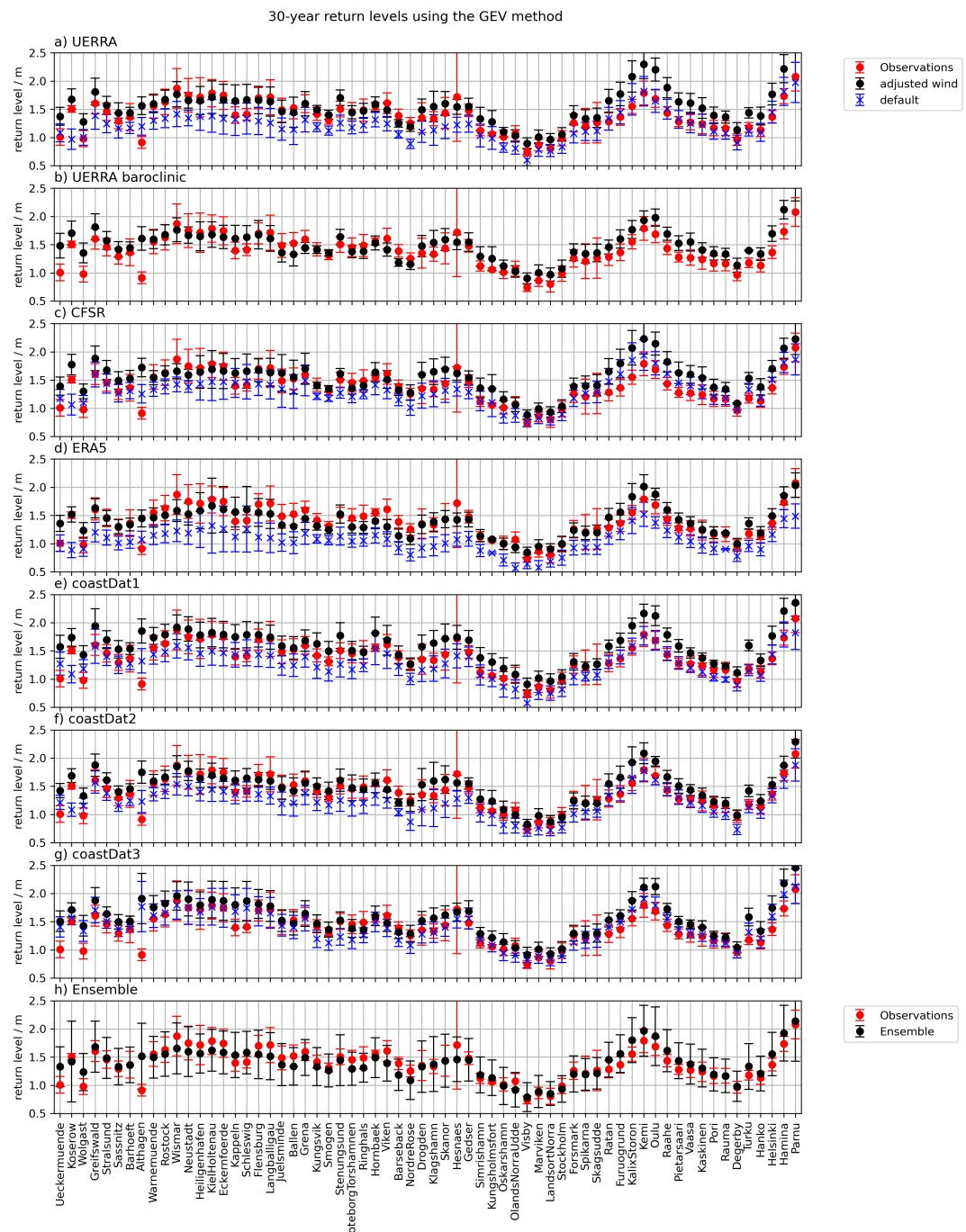

**Figure 4.** Summary of the 30-year return levels using the GEV method for each gauge station and each ensemble member: a)-g) return levels and 95% confidence intervals for each atmospheric forcing and each simulation. h) ensemble mean and ensemble 95% confidence interval. The large 95% confidence interval of the station 'Hesnaes' is explained by a large uncertainty of the fitted shape parameter $\xi$.

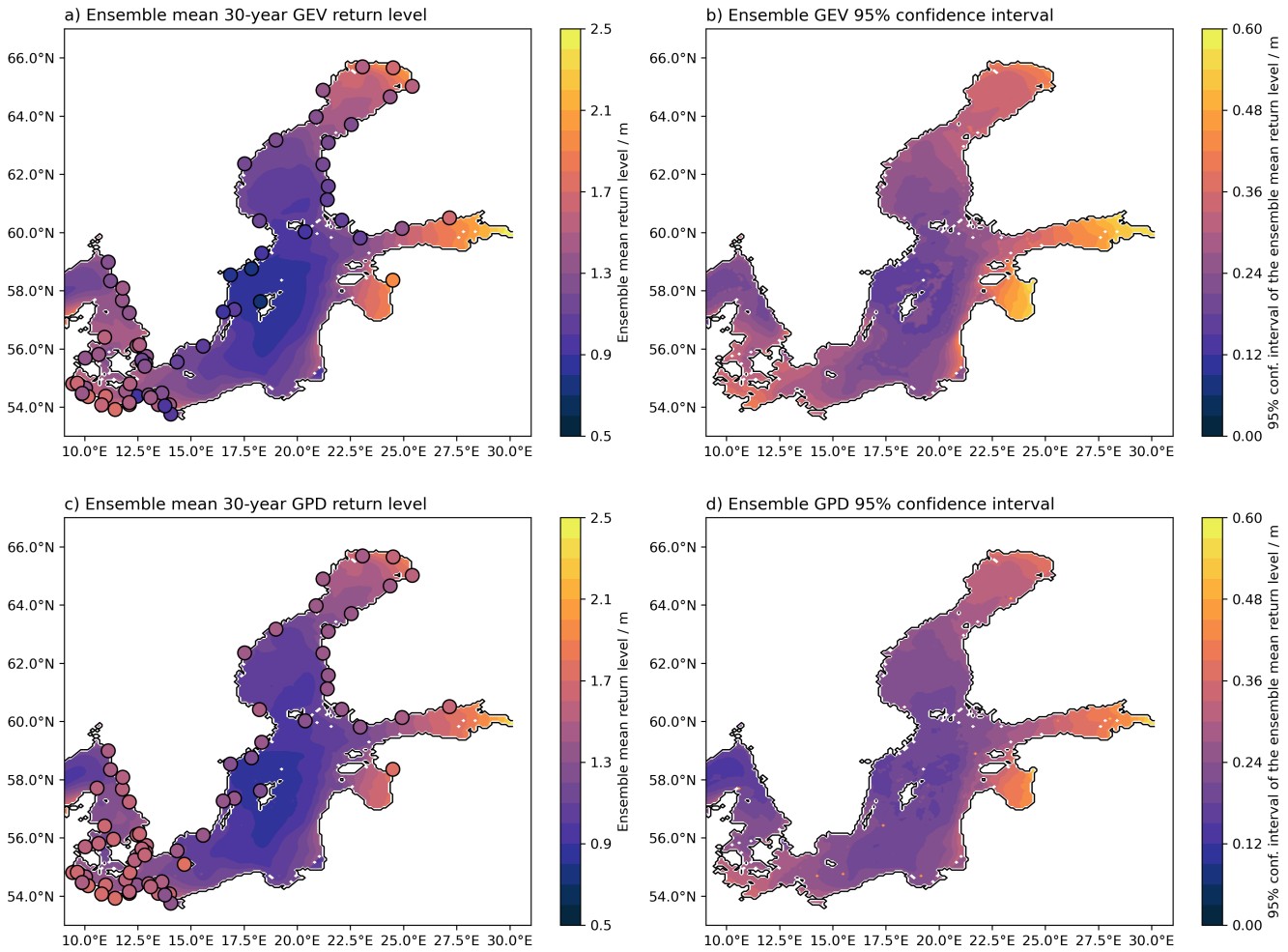

**Figure 5.** Maps of the return levels and the 95% confidence intervals: a) ensemble mean of the 30-year GEV return levels. b) the 95% confidence interval of the GEV return levels. c) ensemble mean of the 30-year GPD return levels. d) the 95% confidence interval of the GPD return levels. In a) and c), the coloured dots indicate the 30-year GEV/GPD return levels of the observations.

incorporates static density effects. Already when comparing the default simulations, the differences between the 30-year return levels of these six simulations can be as high as 60 cm, even though they are all reanalyses.

### 3.2.2 Return levels obtained from the GPD method

Similar to the return levels estimated by the GEV method, we estimated the return levels based on the GPD method for the same period. In the Supplement, we provide plots comparing the estimated return levels of the simulations with the return levels based on the observations: for the station 'Warnemuende' in Fig. S3 and for the 30-year return levels in Fig. S4. In short, comparing the 30-year return levels for each station, the patterns are similar to the GEV results except for the generally lower

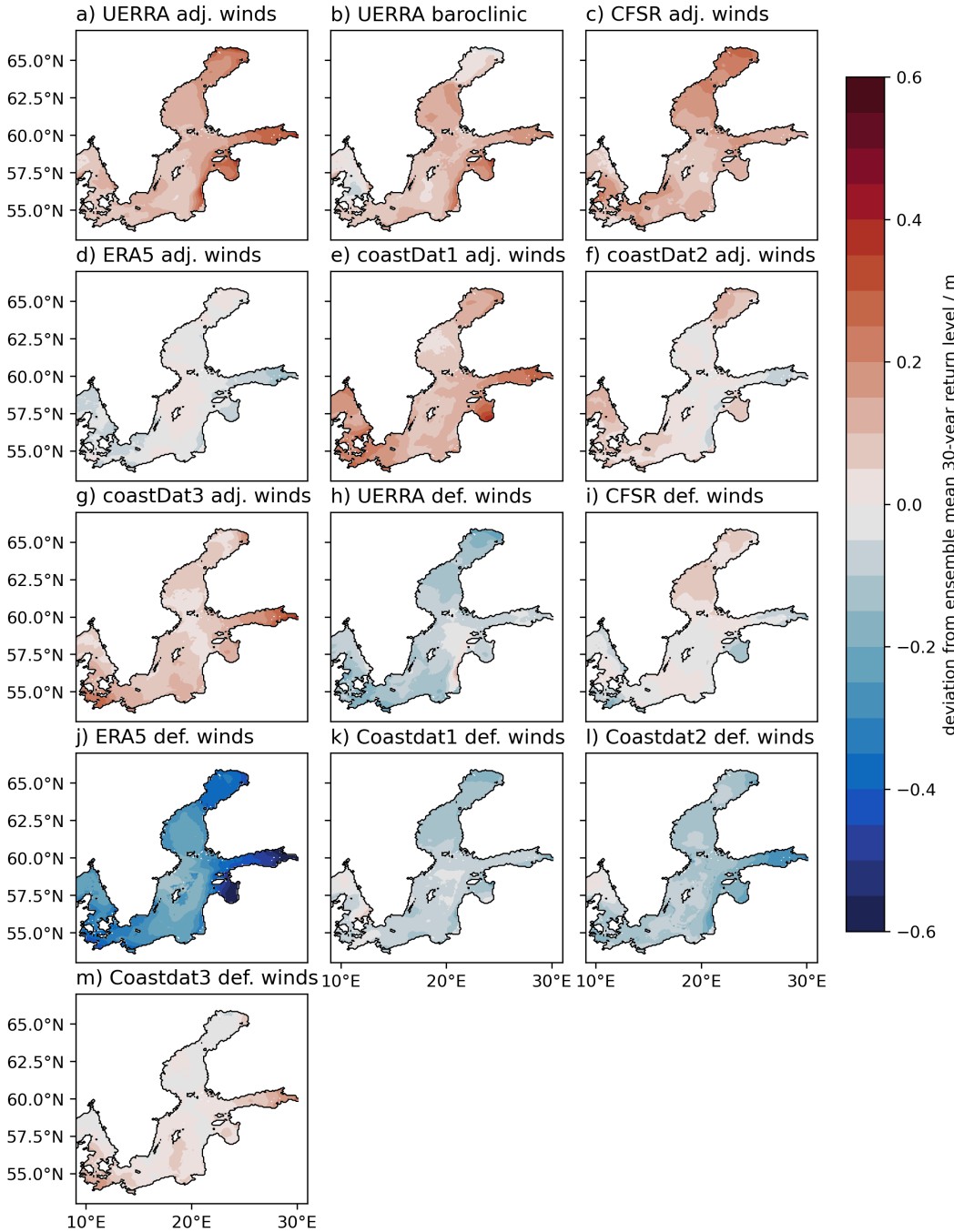

**Figure 6.** Spatial distribution of the 30-year GEV return level deviation from the ensemble mean (see Fig. 5a) for each ensemble member.

estimates for the 30-year return levels. Again, the adjusted wind speed simulations are in better agreement with the observations for the Western Baltic Sea than for the Northern and Eastern Baltic Sea, where the default wind speed simulations are in better agreement with the observations. Overall, based on the observations, the ensemble mean 30-year return levels lie within the 95% confidence intervals of the 30-year return level estimates.

The general pattern of high and low GPD 30-year return levels (Fig. 5c) is very similar to the pattern of high and low GEV 30-year return levels in Fig. 5a. Also, the spatial GPD 95% confidence interval pattern and the values are similar to the GEV 95% confidence interval (Fig. 5b and d). This suggests that the uncertainty within the ensemble is independent of the method used to estimate the return levels. This claim is supported by the deviations of the 30-year GPD return levels from the ensemble mean (Fig. S5 in the Supplement). Each ensemble member shows similar deviations (both in pattern and values) to the GEV

return levels in Fig. 6.

## 3.3  Trends

We have seen in the previous sections that the modelled ESLs differ from observations to a different extents for each atmospheric forcing. Therefore, the estimated 30-year return levels differ significantly within the ensemble, leading to significant uncertainty. Previous studies have shown some trends in storm surge occurrence and heights. Therefore, we look at linear

annual storm season maxima trends within the ensemble period, 1979-2018. For the ensemble mean (Fig. 7) we find a negative mean trend in the Baltic Sea except for the eastern Kattegat and the waters north of Poland. The ensemble mean shows the largest negative trends in the Northern Baltic Sea and the Gulf of Finland. However, the uncertainty within the ensemble is large up to 1.5 mm yr$^{-1}$ (Fig. 7b). Comparing the trends with the observed trends, coloured dots in 7a, we see a general agreement in the trends and in the spatial distribution. However, some gauge stations show deviations in the sign of the trend, e.g. stations

in the Western Baltic Sea. This indicates that details in the trends are not captured by either the ocean model resolution (one nautical mile) or the atmospheric reanalysis data.

    Comparing the trend of each member (Fig. 8), we see an agreement in the spatial pattern of trends as represented by the ensemble mean. However, the size and shape of the areas of positive and negative trends differ, e.g. the extent of the area in the Western Baltic Sea that shows a positive trend varies for each ensemble atmospheric forcing. Furthermore, the actual values

of the trends differ greatly within the different simulations, not only between the adjusted and default wind speeds but even within the sub-sample of default simulations.

## 4  Discussion

From the results of our study, we find that the choice of atmospheric reanalysis significantly impacts how well the ocean model simulates past sea level extremes. Not only are the estimated return levels significantly different depending on the choice of the

atmospheric reanalysis, but the modelled trends in sea level maxima are also affected. One of the reasons is that each simulation forced by a different reanalysis product underestimates or overestimates different ESL events since each real storm system is represented slightly different. Furthermore, each reanalysis has different biases, both spatially and in wind speed distribution

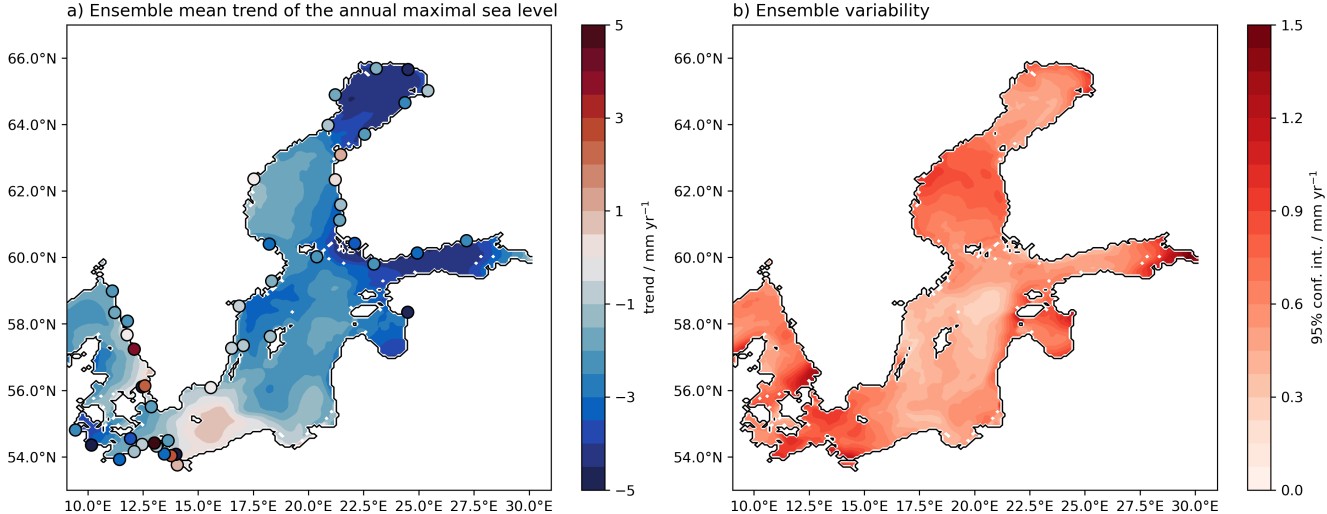

**Figure 7.** The ensemble mean trend in a) and the 95% confidence interval of the mean trend of the annual storm season maxima in b) for the time period from 1979 to 2018. The coloured dots show the trends of the tide gauges covering the same 40 years. Note that the mean sea level trends have been subtracted beforehand.

and direction. Therefore, the variability within the generated ensemble can be quite large, both for the return levels and trends. Nevertheless, the ensemble mean can reproduce observed return levels and observed trends.

The variability within the ensemble for the 30-year return levels can be as high as 60 cm in the Gulf of Finland or the eastern part of the Baltic Proper. Along the western coasts of the Baltic Sea, the 95% confidence values are around 30-40 cm. The 99th percentiles for each default 10 m wind speed ($u_{10}$) dataset from Tab. 1 (Fig. 9) show the variability between the different datasets. ERA5 has the lowest ESLs and has the lowest 99th percentile wind speeds over the Baltic Sea. CFSR shows the highest 99th percentile wind speeds over the Baltic Sea of the ensemble, which translates into the smallest adjustment for wind speeds
of 3%, Tab. 1. The 95% confidence intervals due to the variability within the ensemble are $\sim 2.5\,\mathrm{m\,s^{-1}}$ between the default atmospheric datasets over the entire Baltic Sea. This can result in a difference in the maximal sea levels of approximately

$$|\Delta\eta| = \left|\frac{LC_D\rho_{\mathrm{air}}\left((u_{10}^2-(u_{10}\pm 2.5)^2)\,\mathrm{m^2\,s^{-2}}\right)}{g\rho_0 H}\right| \approx 10-60\,\mathrm{cm}, \tag{7}$$

where $L$ is a distance scale (fetch length) ($\mathcal{O}(L) = 100\,\mathrm{km}$), the drag coefficient $C_D$, which is approximated with 0.001 for simplicity, the density of air $\rho_{\mathrm{air}} \approx 1.25\,\mathrm{kg\,m^{-3}}$, the density of seawater $\rho_0 \approx 1000.0\,\mathrm{kg\,m^{-3}}$, the water depth $H$, and
$g$ the gravitational constant. The sea level difference depends strongly on the water depth and the distance $L$. For $u_{10} = 18.0\,\mathrm{m\,s^{-1}}$, $L = 300\,\mathrm{km}$, and $H = 50\,\mathrm{m}$, the resulting elevation change would be $\sim 43\,\mathrm{cm}$. The resulting sea level change would be even higher for shallower waters or greater distances. This approximation of the effect of the atmospheric variability is in the order of magnitude of the sea level confidence interval of the ensemble. This highlights the importance of correctly representing of high percentile wind speeds, i.e. storms, in the atmospheric reanalysis, to capture the ESLs in the simulations.

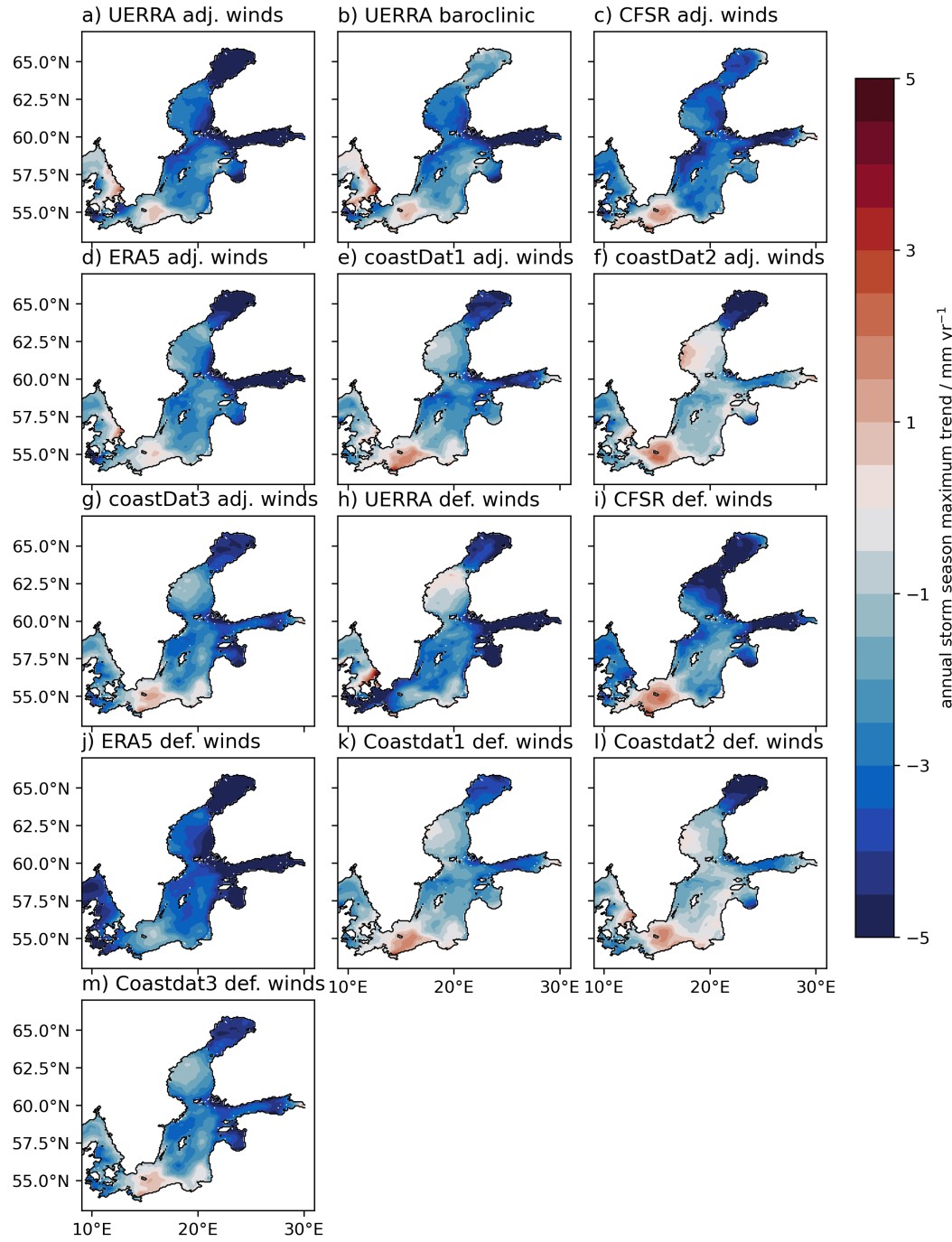

**Figure 8.** Trend in each ensemble members' annual storm season maxima. Note that the mean sea level trends have been subtracted beforehand.

The variability within the ensemble of our study is in agreement with other studies in this regard: Dieterich et al. (2019) discuss the historical period (1961-2005) of a regional climate model for the North Sea and the Baltic Sea to study the variability within an ensemble of six members (six different global model boundary conditions for their regional general circulation model). They find uncertainties between 20-40 cm added by the choice of the global circulation model boundary conditions as the largest uncertainty for the 100-year return levels. Eelsalu et al. (2014) show for their small three-member ensemble for the Estonian

coast that uncertainties for 15-year return levels are in the range of about 20 cm. However, for longer return periods, their spread grows significantly to 60 cm and more. In general, the errors grow with longer return periods, as does the variability of our ensemble. To reduce the uncertainty, we believe a large regional ensemble is needed.

The variability of the return levels within the ensemble is independent of the choice of chosen extreme value distribution. Our results show similar confidence intervals within the ensemble for both the GEV and GPD methods: both show similar

spatial patterns and similar values. However, their absolute values for the return levels differ, especially for the short and very long return periods. Based on the annual maxima of the storm season, the GEV method yields higher 30-year return levels than those obtained from the GPD method. We do not want to discuss the pros and cons of the methods. We refer to Arns et al. (2013) for detailed discussion of this. However, the difference in the return levels of the two methods for the 30-year return levels shown in this study is smaller or in the range of variability within the ensemble. This indicates that future efforts should

be made to minimise the uncertainty. ESLs are the superposition of many sea level processes: preconditioning, storm surges, and standing waves (seiches). For example, since preconditioning can increase the mean sea level in the Baltic Sea on weekly time scales, the total sea level during a storm surge can be increased. These two (and more) compounding mechanisms follow different statistical distributions (e.g. Suursaar and Sooäär, 2007) which can be extracted by separating the time series into different temporal components (Soomere et al., 2015). Our analysis neglects this point as the GEV and GPD methods assume

a single statistical distribution which still led to good fits. Furthermore, the mentioned mechanisms are interacting non-linearly (Arns et al., 2020). Disentangling these two (and more) compounding processes (Soomere and Pindsoo, 2016; Pindsoo and Soomere, 2020), of this ensemble remains a matter for a future study.

We have shown with the generated ensemble that the trend of the annual storm season maxima is different for each ensemble member, but still shows some agreement in the general spatial pattern. Since each storm is captured differently in each

atmospheric dataset, this influences the trends of the ESLs. Fig. 10 shows the linear trends of the annual 99th percentile wind speeds for each dataset and the ensemble mean (interpolated to the UERRA grid). The datasets agree on negative trends for the southern, western, and central Baltic Sea for the hind-cast period. We found a positive trend for most datasets for the northern and eastern parts. CoastDat2 deviates from this general pattern with very small trends in the Western Baltic Sea and negative trends in the north. Since the CFSR-member is forced with both CFSR and CFSv2 no consistent model is used and the trends

shown are expected to be mostly due to the change in the model configuration. An estimate of how the wind trends can be translated into sea level trends can be made using the derivative of equation (7):

$$|\partial_t \eta| = \left| \frac{2 L C_D \rho_{\mathrm{air}} u_{10} \partial_t u_{10}}{g \rho_0 H} \right| \approx 0.1 - 1.0 \, \mathrm{mm/yr}, \tag{8}$$

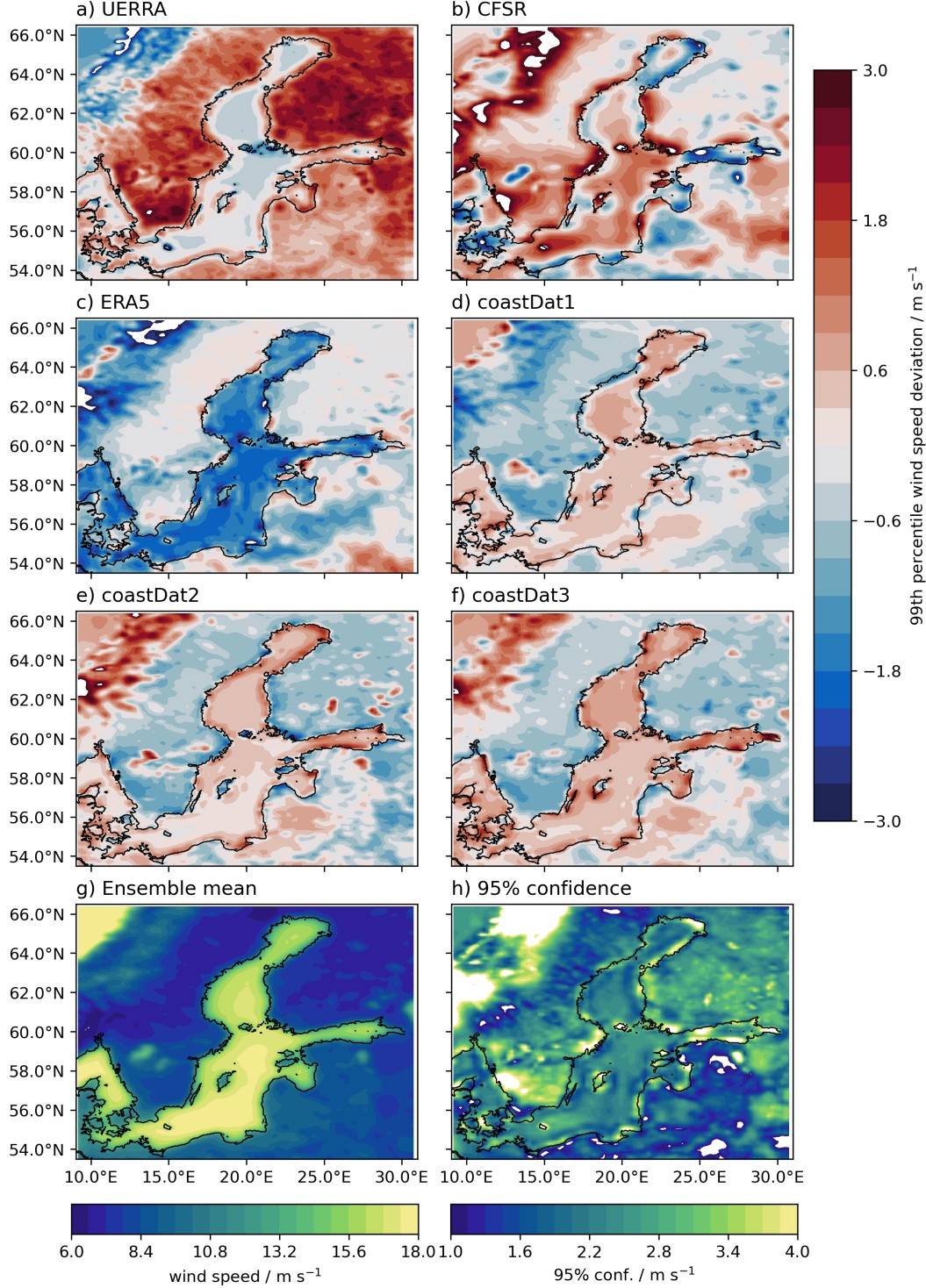

**Figure 9.** Deviations of the 99th percentiles of the 10 m wind speeds from their mean for the meteorological forcings used to generate the ensemble. The mean in g) and the standard deviation in h) are computed by interpolating the individual percentiles of each forcing onto the grid of the UERRA dataset.

where the value $\partial_t X$ denotes the slope of the trend line. For $u_{10} = 18.0\,\mathrm{m\,s^{-1}}$, $L = 300\,\mathrm{km}$, and $H = 50\,\mathrm{m}$, and $\partial_t u_{10} = 0.03\,\mathrm{m\,s^{-1}\,yr^{-1}}$ the resulting change in elevation would be $\sim 0.7\,\mathrm{mm\,yr^{-1}}$. However, the ensemble trends of the sea level maxima are higher, indicating that directional changes and preconditioning must play a critical role. However, this is beyond the scope of this study. Furthermore, if we compare the trends of the ensemble mean to the results of Soomere and Pindsoo (2016) and Pindsoo and Soomere (2020), we find an apparent discrepancy. Whereas Soomere and Pindsoo (2016) and Pindsoo and Soomere (2020) find a clear positive trend of up to $10\,\mathrm{mm\,yr^{-1}}$, the ensemble of this study shows a negative trend. They further show that preconditioning is responsible for up to $4-5\,\mathrm{mm\,yr^{-1}}$ of the total trend, showing that preconditioning indeed plays a crucial role. It should also be noted that these trends are sensitive to the considered period since the natural variability of the Baltic Sea region includes oscillations in the order of decades due to the North Atlantic Oscillation (NAO) and the Atlantic Multidecadal Oscillation (AMO) (also named Atlantic Multidecadal Variability, AMV) (Meier et al., 2022b, and references therein). The considered time period of Soomere and Pindsoo (2016) and Pindsoo and Soomere (2020) is from 1961 - 2005 with 6-hourly model output. We compared the trends for tide gauge station which go back to 1961 for both time periods, 1979-2018 and 1961-2005, Tab. S2. The results show that both time periods lead to different trends, even changing the sign of the trend. Many of these stations show a positive trend for 1961-2005 and negative trends for 1979-2018. For example, stations in the Western Baltic Sea show a positive trend up to $4\,\mathrm{mm\,yr^{-1}}$ which is in agreement with the previous studies. However, this is not the case for 'Parnu' which still shows a negative trend. Therefore, the discrepancy between this study and previous studies can be only partly attributed to the different time periods. This hints that the variability of ESLs to the NAO and AMO/AMV should be revisited. Furthermore, the 6-hourly output of Soomere and Pindsoo (2016) and Pindsoo and Soomere (2020) could introduce some aliasing since peak water levels may not be captured, compare to e.g. to Kiesel et al. (2023) for mean time series of ESLs in the Western Baltic Sea.

Our ensemble has a record length of 40 years. The 100-year or 200-year return levels are often used as protection targets for coastal protection planning. Therefore, an extrapolation of the data leads to significant uncertainty ranges, see e.g. Fig. S2 and S3 in the Supplement, where the confidence intervals grow significantly when extrapolating, especially for the GEV method. To reduce the uncertainties, longer time series would be necessary to estimate the long return periods reliably. For the Baltic Sea, very long ($> 100$ years) time series exist and have been used to study the large return levels (e.g. Suursaar and Sooäär, 2007). However, our results show that climate change makes reliable estimates of the highest return levels for the current and future climate states based on small ensembles or single simulations complicated and uncertain, see also Muis et al. (2022). It is expected that the underlying statistical distribution will change with climate change. Therefore, the non-stationary extreme value statistics have been used in other studies, e.g., Kudryavtseva et al. (2021) and temporal trends in ESLs have been studied, e.g., with quantile regression (Barbosa, 2008). Disentangling these changes in statistical extreme value distributions is only possible with numerical models. However, much larger ensembles are needed to separate ensemble variability from changes due to climate change. Currently, the ensemble variability for climate scenarios for the European coasts is in the order of magnitude of the expected differences between low and high RCP scenarios (e.g. Vousdoukas et al., 2017). In addition, the large natural variability in the Baltic Sea regions makes this disentanglement even more complicated in this region.

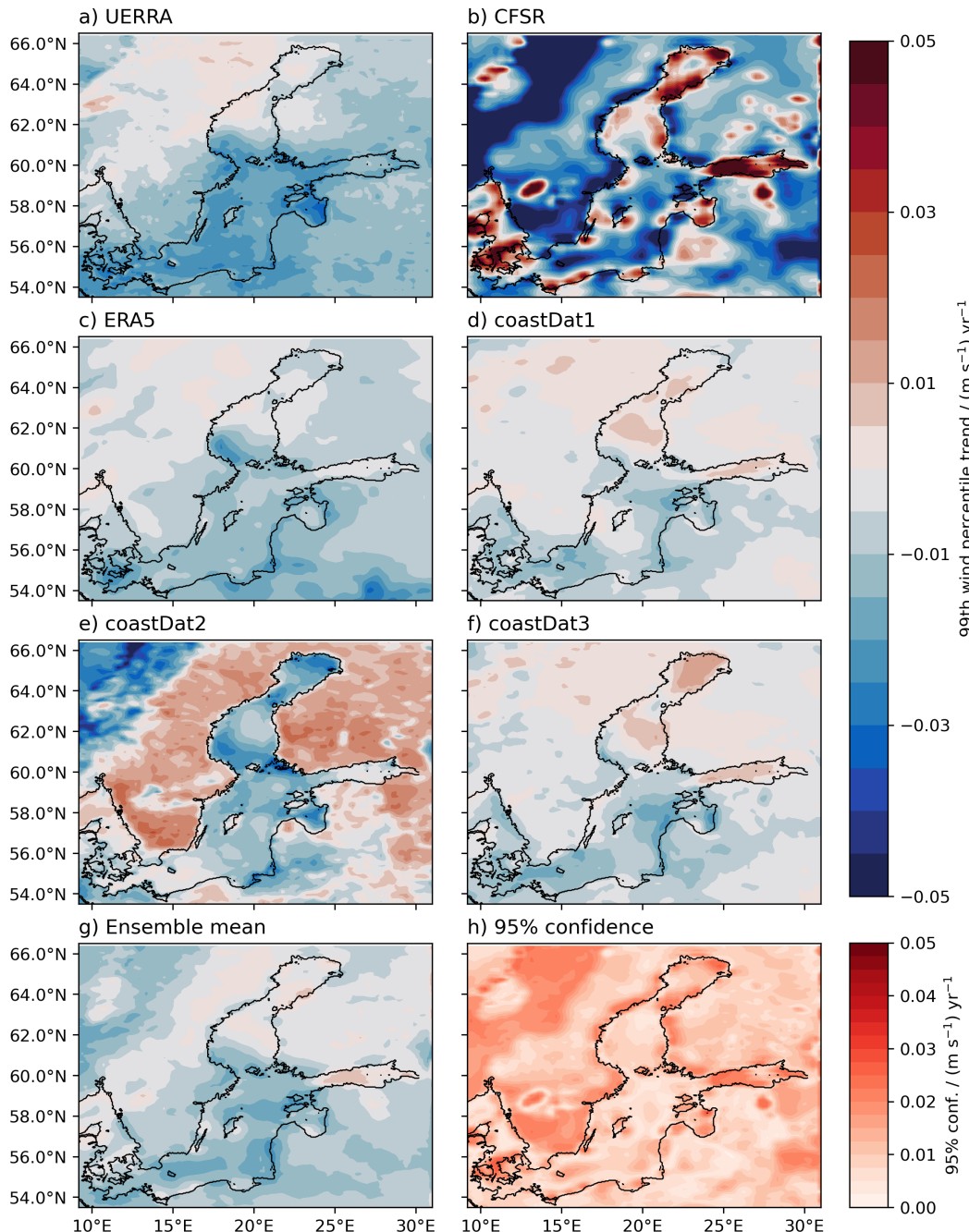

**Figure 10.** Trend of the annual 99th percentiles of 10 m wind speeds for the meteorological forcings used to generate the ensemble.

To reduce model bias (Section 3.1), we increased the wind speeds by a constant factor which reduced the mean bias of ESLs. However, the results suggest that the wind factor should be varied spatially to account for regional biases in ESLs in the simulations. The main negative model bias in our, but also previous studies, is found in the ESLs of the Western Baltic Sea. In early model studies the bias was because of coarse resolution which allowed water to exit through the Danish Straits during ESL events (e.g. Meier et al., 2004). Furthermore, local convergence effects need high model resolution to be correctly reproduced. But also in high-resolution hydrodynamic models negative biases in ESLs are found: 1 km resolution in Gräwe and Burchard (2012), 200 m resolution in Kiesel et al. (2023). Therefore, biases can be attributed to wind speed biases (e.g. Gräwe and Burchard, 2012). In the present study, we found the ESL biases to be caused by biases in the atmospheric datasets or how the ocean model translates the forcing into sea level changes. Applying better corrections is not a trivial undertaking and would certainly further disturb the physical consistency. Adjusting wind speeds may help with the biases in the ESLs, but it also has implications for general hydrodynamics, which is not the focus of this study. Because we use vertically integrated Reynolds averaged Navier-Stokes equations, we do not consider changes in the surface mixed layer depths, or upwelling and downwelling. Nevertheless, these would be strongly modified by the adjusted wind speeds and may lead to over- or underestimation of these. This would have consequences for the general dynamics, e.g. stratification or the total salt import by Major Baltic inflows. One possible way of tackling this problem could be adjoint/inverse models (Errico, 1997) which can be used to calibrate coastal ocean input fields like bottom friction (Kärnä et al., 2023) or in this case wind speed factors. However, this does not solve the underlying issue which we believe is found in the resolution of the atmospheric models over the Western Baltic Sea. Compared to the size of the Western Baltic Sea, the number of grid cells in the atmosphere is probably not high enough to resolve the winds properly.

There exist many formulations, i.e. polynomial fits, of the drag coefficient which translate wind speeds into winds stresses (e.g. Kara et al., 2000; Large and Yeager, 2009). Here we have used the formulation of Kara et al. (2000). Using a different formulation, e.g. one that yields a larger drag coefficient for high wind speeds, could have improved the representation of ESLs and potentially decreased the wind factors we applied. Since there exist many formulations, each formulation would lead to different ESLs, thus introducing a source for variability. The different formulations depend on the geographic locations of the underlying data which is fitted, e.g. offshore versus onshore (Smith et al., 1992). The formulation we have used (Kara et al., 2000) is based on observations made in the Arabian Sea. One could argue that this formulation may not suitable for the Baltic Sea. However, we expect for this study that the variability of the atmospheric datasets, Figs. 9 and 10, to be much larger than the differences in ESLs due to the drag coefficient formulations. Still, the exploration of the effect of different drag coefficients formulations on ESLs in the Baltic Sea should be carried out in a future study. A calibrated drag coefficient formulation using an adjoint model could be a possible solution (Peng et al., 2013). However, depending on the wind input fields, different best fits have to be expected.

We have increased the ensemble spread by including the default and adjusted wind speed simulations. We could have either used only the default wind speed simulations and accepted the negative biases in the Western Baltic Sea, thus reducing the ensemble spread, or we could have included them to improve the ensemble mean for that region, with the drawback of larger uncertainties. Ultimately, uncertainties are shifted from one problem to another and would not be minimised.

## 5 Summary and Conclusions

We show that modelling extreme sea levels is difficult and relies heavily on correctly representing storms in the atmospheric forcing data. We derive and provide a 13-member hind-cast ensemble focusing on extreme sea levels, covering the period from 1979 to 2018. We constructed the ensemble using present state-of-the-art reanalysis datasets for the Baltic Sea region. We have explicitly not considered long-term changes in mean sea level. The ensemble mean can reproduce similar extreme value distributions compared to tide gauges around the Baltic Sea, both for the Generalised Extreme Value distribution (GEV) and the Generalised Pareto distribution (GPD). However, spatially heterogeneous biases can be expected depending on the choice of atmospheric forcing. We find a large variability within the ensemble: up to $60\,\mathrm{cm}$ (95% confidence interval) within the ensemble for the 30-year return levels and trend variability up to $1.5\,\mathrm{mm\,yr^{-1}}$ for the annual maximum sea level. This variability is entirely due to the different atmospheric representations of storms and how these translate into the sea level computation in the ocean model. One approach trying to minimise this uncertainty can be much larger ensembles. Hence, we invite other scientists to incorporate our numerical simulations and results into future ensembles to increase the number of ensemble members. Initiatives such as the Baltic Sea Model Intercomparison Project (BMIP, Gröger et al. (2022), https://www.baltic.earth/working_groups/model_intercomparison/index.php.en) should therefore be expanded in the future to build a large regional ensemble with multiple forcing combinations.

Since the trends of annual maximum sea levels are sensitive to the time period considered and the trends of this study differ from previously published studies (e.g. Soomere and Pindsoo, 2016; Pindsoo and Soomere, 2020), we believe that the dependence of ESLs on the NAO and AMO/AMV should be reconsidered. Furthermore, the contributions of preconditioning, seiches, and surges to ESLs in the context of climate modes need to be disentangled.

*Code and data availability.* The results, a frozen GETM version, and the analysis code can be found here: https://doi.org/10.5281/zenodo.8340649. The model output can be accessed here: https://thredds-iow.io-warnemuende.de/thredds/catalogs/publications/lorenz/catalog_Lorenz_Graewe_2023.html. The atmospheric forcings can be accessed from following links: UERRA https://doi.org/10.24381/cds.44ec8078, ERA5 https://doi.org/10.24381/cds.adbb2d47, CFSR https://www.ncei.noaa.gov/products/weather-climate-models/climate-forecast-system, coast-Dat1, coastDat2, coastDat3: personal communication from HEREON, general website https://www.coastdat.de/.

*Author contributions.* ML and UG designed the study together. UG created the model setups. ML performed the simulations, did the analyses, and wrote the first draft of the manuscript with input from UG.

*Competing interests.* The authors indicate no competing interests.

*Acknowledgements.* The authors thank Tarmo Soomere and one anonymous reviewer for their constructive feedback. This research is part of the ECAS-BALTIC project: Strategies of ecosystem-friendly coastal protection and ecosystem-supporting coastal adaptation for the German Baltic Sea Coast. The project is funded by the German Federal Ministry of Education and Research (BMBF, funding code 03F0860H). All simulations were performed on clusters of the German National High Performance Computing Alliance (NHR). Figures in this paper use the colourmaps of the cmocean package (Thyng et al., 2016).

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
