# Peer review of "Uncertainties and discrepancies in the representation of recent storm surges in a non-tidal semi-enclosed basin: a hind-cast ensemble for the Baltic Sea"

_EGUsphere, 2023_

## Author Response (AR1)

**Author response to reviewer 1 comments**

Marvin Lorenz and Ulf Gräwe

September 8, 2023

We thank both referees, Tarmo Soomere and one anonymous reviewer, for their detailed comments and thoughts. Our responses to reviewer 1 are given below.

> "This is a nice piece of study that addresses mismatches of water level extremes in the Baltic Sea simulated using different high-quality modelled wind data sets. Although such mismatches are known for some time, it is the first systematic analysis of the discrepancies, associated uncertainties and ways towards a reasonable forecast of extreme water levels for moderately long (30 years) return periods based on the application of an ensemble of hindcasts.
>
> It is encouraging that the average of the resulting ensemble represents well return periods extracted from modelled data. The analysis of spatial variations in the trends of annual (storm-season) water level extremes (both modelled and extracted from measurements) is also interesting and valuable even though there are also major uncertainties and discrepancies between the ensemble members.
>
> The analysis is sound and generally well presented. The use of English is appropriate. The presentation still needs some polishing as the text contains numerous examples of unnecessary field-specific jargon that would be nice to remove (unless specifically justified, e.g., for brevity). In particular, the use of default wind speed simulations to denote the unadjusted results of such simulations is misleading. Consider using unadjusted or similar instead.
>
> The results are generally presented in proper context, so I have only a few remarks (see below). The most intriguing conjecture is that there has been effectively no increase in the Baltic Sea water level extremes over last decades even though the average water level has been increasing. One the one hand, this outcome is basically consistent with the general perception of the Baltex/Baltic Earth community that wind speed extremes and the number of strong storms have not substantially increased in the Baltic Sea region. On the other hand, it calls for further analysis of the reaction of Baltic Sea water masses to changing wind forcing both in terms of waves and (extreme) water levels as several analyses (e.g., Jaagus, J., Suursaar, ., 2013. Long-term storminess and sea level variations on the Estonian coast of the Baltic Sea in relation to large-scale atmospheric circulation. Est. J. Earth Sci. 62, 7392. https://doi.org/10.3176/earth.2013.07) have identified rapid increase in water level extremes in the eastern Baltic Sea."

Thank you for your feedback. The word "default" to us has the meaning of a pre-designed value or setting. This is the standard use of the word in computer sciences. However, you are right that its original meaning is something like "fail / failure", mainly governing financial things like loan payments. Since the word is the common word to describe the pre-designed settings, we decided not to change it.

> It might be mentioned that the most extreme water levels in the Baltic Sea often represent a different population of extremes (because of preconditioning) and thus the classic approaches such as the GEV technique (that presumes the presence of one population) do not necessarily work as noted in (Suursaar and Soor, 2007) and mentioned several times later. This specific population is visible in Figures S1 and S2 as a few highest extremes that do not match the theoretical curve.

This is a good point. We added to the discussion of the return levels: "ESLs are the superposition of many sea level processes: preconditioning, storm surges, and standing waves (seiches). For example, since preconditioning can increase the mean sea level in the Baltic Sea on weekly time scales, the total sea level during a storm surge can be increased. These two (and more) compounding mechanisms follow different statistical distributions (e.g. Suursaar and Sooäär, 2007). Our analysis neglects this point as the GEV and GPD methods assume a single statistical distribution which still led to good fits. Furthermore, the mentioned mechanisms are interacting non-linearly (Arns et al., 2020). Disentangling these two (and more) compounding processes of this ensemble remains a matter for a future study." We added to the conclusions: "Furthermore, the contributions of preconditioning, seiches, and surges to ESLs in the context of climate modes need to be disentangled."

> Line 156: On average, the simulations underestimate the maximum water levels and line 161 However, the wind speed increase is necessary to capture the correct ESLs in the Western Baltic Sea are very strong results/claims. Please comment whether this is a typical issue in other studies and/or provide supporting references.

In previous studies, the coarse resolution was found to increase the volumetric exchange with the North Sea during storm surges, thus the water cannot pile up high enough in the Western Baltic Sea (Meier et al., 2004). Furthermore, local convergence due to the coastlines could be underestimated in our model (Gräwe and Burchard, 2012). However, even in a high-resolution 200m simulation of the Western Baltic Sea, the ESLs were too low on average (Kiesel et al., 2023). Thus, we believe the main bias in the Western Baltic Sea originates from the atmospheric datasets, see also Gräwe and Burchard (2012). We added new sentences to the paragraph of the discussion on the biases of the model and the wind speed factor correction: "In early model studies the bias was because of coarse resolution which allowed water to exit through the Danish Straits during ESL events (e.g. Meier et al., 2004). Furthermore, local convergence effects need high model resolution to be correctly reproduced. But also in high-resolution hydrodynamic models negative biases in ESLs are found: 1 km resolution in Gräwe and Burchard (2012), 200 m resolution in

Kiesel et al. (2023). Therefore, biases can be attributed to wind speed biases (e.g. Gräwe and Burchard, 2012). "

"However, this does not solve the underlying issue which we believe is found in the resolution of the atmospheric models over the Western Baltic Sea. Compared to the size of the Western Baltic Sea, the number of grid cells in the atmosphere is probably not high enough to resolve the winds properly."

> The obtained predominantly negative trends of annual sea level maxima obtained in Section 3.3 radically differ from large positive ones reported in (Pindsoo and Soomere, 2016) // (Soomere and Pindsoo, 2020) for the entire Baltic Sea. Could the difference stem from different time periods? Or different forcing and ocean models? Or the adequacy of replication of water exchange via the Danish straits?

Yes, this result was also surprising to us. But since the observations showed similar trend and a similar spatial pattern, we are confident that the trends within the time period are robust.
The main differences to Soomere and Pindsoo (2016) despite the model data are the resolution of the sea level output 6-hourly in Soomere and Pindsoo (2016) and the considered time period from 1961-2005. We compared the trends for 1979-2018 and 1961-2005 for tide gauge which cover both periods, Tab. 1 which is Tab. S2 in the revised supplement. We find that indeed the time period does change the sign of the trends for stations in the Western Baltic Sea, but there are still disagreements with previous studies, e.g. for 'Pärnu. This hints that the ESL dependency on AMV and NAO should be revisited in the future. For the western Baltic Sea, the temporal resolution of 6-hourly output may introduce some aliasing in maximum water levels since peak water levels are usually quite short, compare Kiesel et al. (2023). Since most of the atmospheric datasets show negative trends for the high wind speed percentiles, we believe that this trend translates into the maximum sea level trends.
We do not believe that water exchange via the Danish Straits plays a major role in this since the trends seem to be very dependent on the chosen time period as mentioned before. We added and changed the respective paragraph in the discussion:
"Furthermore, if we compare the trends of the ensemble mean to the results of Soomere and Pindsoo (2016) and Pindsoo and Soomere (2020), we find an apparent discrepancy. Whereas Soomere and Pindsoo (2016) and Pindsoo and Soomere (2020) find a clear positive trend of up to $10 \, \mathrm{mm \, yr^{-1}}$, the ensemble of this study shows a negative trend. It should also be noted that these trends are sensitive to the considered period since the natural variability of the Baltic Sea region includes oscillations in the order of decades due to the North Atlantic Oscillation (NAO) and the Atlantic Multidecadal Oscillation (AMO) (also named Atlantic Multidecadal Variability, AMV) (Meier et al., 2022, and references therein). The considered time period of Soomere and Pindsoo (2016) and Pindsoo and Soomere (2020) is from 1961 - 2005 with 6-hourly model output. We compared the trends for tide gauge station which go back to 1961 for both time periods, 1979-2018 and 1961-2005, Tab. S2. The results show that both time periods lead to different trends, even

Table 1: Comparison of the trends of annual storm season maxima for to different time slices: 1979-2018 and 1961-2005.

| station | trend 1979-2018 / mm yr$^{-1}$ | trend 1961-2005 / mm yr$^{-1}$ |
|---|---|---|
| Hornbaek | -3.35 | -2.10 |
| Klagshamn | -1.32 | 1.99 |
| Flensburg | -2.09 | -0.58 |
| Furuogrund | -0.25 | -0.35 |
| Gedser | -3.37 | 2.26 |
| Kungsholmsfort | 0.28 | 1.02 |
| Parnu | -5.92 | -1.40 |
| Ratan | -0.40 | 0.72 |
| Sassnitz | -2.01 | 1.06 |
| Smogen | -0.69 | 3.29 |
| Stockholm | -0.94 | -1.31 |
| Warnemuende | -1.07 | 0.76 |
| Wismar | -3.33 | 3.83 |

changing the sign of the trend. Many of these stations show a positive trend for 1961-2005 and negative trends for 1979-2018. For example, stations in the Western Baltic Sea show a positive trend up to 4 mm yr$^{-1}$ which is in agreement with the previous studies. However, this is not the case for 'Parnu' which still shows a negative trend. Therefore, the discrepancy between this study and previous studies can be only partly attributed to the different time periods. This hints that the variability of ESLs to the NAO and AMO/AMV should be revisited. Furthermore, the 6-hourly output of Soomere and Pindsoo (2016) and Pindsoo and Soomere (2020) could introduce some aliasing since peak water levels may not be captured, compare to e.g. to Kiesel et al. (2023) for mean time series of ESLs in the Western Baltic Sea. "

We added to the conclusions:

"Since the trends of annual maximum sea levels are sensitive to the time period considered and the trends of this study differ from previously published studies (e.g. Soomere and Pindsoo, 2016; Pindsoo and Soomere, 2020), we believe that the dependence of ESLs on the NAO and AMO/AMV should be reconsidered."

The Discussion section provides some claims that seem to be overly simplified reflections of the results. For example, the sentence on page 16, line 234235 Nevertheless, the ensemble agrees well with observed return levels and trends might be augmented to mention that in quite many locations the trends from measured data have even different signs compared to trends evaluated from simulations.

We reworked many parts of the discussion, so the raised point should be resolved.

Minor comments:

Page 2, lines 2526: the meaning of the sentence However, the representation of ESLs in numerical models adds another source of uncertainty remains unclear (or at least ambiguous); please explain what is meant.

Rephrased to "However, the representation of ESLs in numerical models is depending on the representation of storms in the atmospheric dataset, adding another source of uncertainty."

Line 41: please indicate the year for reference (IPCC).

Added the year 2021.

Line 43: their Tab. 2. probably it is meant Table 2 of (Bamber et al., 2019); please make the link clear.

Rephrased to "Considering ice sheet dynamics in more detail, Bamber et al. (2019) list in their Tab. 2 possible ranges from 69 to 111 cm (median), 49-98 to 79-174 cm (17-83%), and 36-126 to 62-238 cm (5-95%)."

Line 4445: the claim With this rise, the exchange of water masses with the North Sea is expected to increase, which could affect preconditioning is not fully justified. I guess that after water level rise in the North Atlantic there will be simply a new balance of water masses of the North Sea and Baltic Sea. However, as the Danish straits and the Baltic Sea will be deeper, the wind-driven slope of the Baltic Sea water could be smaller than today and thus the water exchange in the context of the current study may even be reduced. This change, however, will affect preconditioning but as mentioned does not necessarily enhance this process. If my reasoning is not correct, please provide additional evidence or arguments to support the claim.

We removed the sentence.

Line 48: singular seems not appropriate as this word has clear meaning in mathematics and physics; consider saying exceptional or similar.

Changed to "exceptional".

Lines 4749: the estimate needs a supporting reference.

We removed the sentence.

Line 51: must be therein.

Changed.

Line 51: probably spatially heterogeneous is meant and not temporal heterogeneity.

We added the word "spatially" for clarity.

Line 54: please adjust wording for clarity.

Rephrased to: "Mean sea-level rise in the Baltic Sea also depends on atmospheric changes, such as atmospheric pressure, wind speed, and wind direction (Gräwe et al., 2019)."

Line 5556: return periods  become more frequent is nonsensical; please reformulate.

Rephrased to: "In general, sea-level rise shifts the ESLs (relative to present-day mean sea level) to higher base levels, and thus high ESLs become more frequent (Wahl et al., 2017)."

Lines 5657: This inhomogeneity in sea-level rise and atmospheric changes has led to more frequent and longer ESL events does not make much sense unless an area is indicated.

We rephrase to "Present sea-level rise and atmospheric changes have already led to more frequent and longer ESL events (Wolski and Wiśniewski, 2021)."

Page 3, Line 60: shifts were detected in (Kudryavtseva et al., 2021) for the Gulf of Riga while most changes were basically smooth on the Latvian seashore of Baltic proper.

Rephrased to "For the Gulf of Riga, Kudryavtseva et al. (2021) show that the shape parameter has changed significantly and often abruptly in the past."

Line 64: style of references is inconsistent.

Fixed the style.

Line 77: says On the other hand  but One the one hand  is missing.

It seems to be a one-handed pirate. We omitted "On the other hand".

Line 88: should be Baltic Sea or Baltic region or similar.

Rephrased to "Baltic Sea".

Page 4, Figure 1: there is no need to use expanded West-East scale.

The stretch is due to the chosen projection.

Lines 101103: I agree with both claims (and However should be removed) but you lose interaction of tides with water exchange through the Danish straits. Please either provide an explanation why this is negligible or mention this assumption when describing limitations of the study.

"However" is removed. We do not expect significant changes in the barotropic water exchange in the Danish Straits. The long-term filling of the Baltic Sea is known to be of atmosphere driven nature, so we do not expect any effects of the tides on those time scales. For the short time scales of the tides the Danish Straits act as a low-pass filter and are thus negligible (Gräwe and Burchard, 2012).

Lines 108110 and Page 5, caption to Table 1: please minimize repeated information.

We shortened the caption of Tab. 1.

Line 114: land uplift Was land uplift included in the GETM model? Or how/elsewhere?

No, land uplift it is not considered by the model. The sentence is indeed confusing, thus we removed "land uplift" from the sentence.

Lines 117119 and caption to Fig. 1 provide repeated information; please adjust for clarity and brevity.

We shortened the caption of Fig. 1, but we kept Lines 177-119 unchanged.

Line 125: it is recommended to refer here also to (Kudryavtseva et al., 2018. Non-stationary modeling of trends in extreme water level changes along the Baltic Sea coast. Journal of Coastal Research, Special Issue No. 85, 586590, doi: 10.2112/SI85-118.1) that signals non-stationarity of GEV parameters for extreme water levels in several locations of the Baltic Sea while (Kudryavtseva et al., 2021) focuses on Latvian waters.

Reference is added.

Page 6, Figure 2: consider indicating also original names of water level gauge locations (e.g., Warnemnde, Prnu).

We prefer to avoid using "ä" etc. in the station names.

Page 7, line 129: using the time series of annual storm season (July to June) maxima, such as block maxima is a good idea; however, there should be either a direct justification of this choice or a reference to, e.g., (Mnnikus et al., 2020. Variations in the mean, seasonal and extreme water level on the Latvian coast, the eastern Baltic Sea, during 19612018. Estuarine Coastal and Shelf Science, 245, Art. No. 106827, https://doi.org/10.1016/j.ecss.2020.106827) where this justification is provided. The issue is fundamental as water level maxima over calendar years are not necessarily independent owing to preconditioning, and then the GEV technique is not applicable. Also, remove , such.

Reference is added, "such" is removed. The justification is that we want to capture the winter storm season within one block.

Line 133: consider replacing subfamilies are defined by saying that the GEV is reduced to one of the following distributions depending on the sign of the shape parameter.

rephrased to "The shape parameter $\xi$ governs the tail of the distribution and depending on its value the distribution reduces to:"

Eqs. (3), (4): there is no need to introduce $\alpha$ but if you do so, please give the link to $\xi$ (even though this link is basically trivial).

We replaced $\alpha$ with $\xi$ for these two equations.

Line 146: I guess that a Python script is used also for this fitting.

Yes, it is mentioned now in the text.

Page 8, Lines 152153: use simply ESLs (Fig. 3), where and add (Fig. 1) to the end of sentence, so that the reader is directed to the relevant map.

Rephrased to: "Since we are interested in the ESLs, we compare the observed ESLs for each station with the modelled ESLs (Fig. 3), where the stations are sorted in a clockwise direction along the Baltic Sea coast, starting from the Western Baltic Sea (Fig. 1)."

Line 155: the default wind speed simulations is nonsensical as mentioned above, most likely something like the simulations forced with wind properties from the atmospheric models or similar is meant.

See the first point.

Line 156: On average, the simulations underestimate the maximum water levels is a very strong result/claim as mentioned above. Please comment whether this is a typical issue in other studies. For example, (Pindsoo and Soomere, 2020) who use another model and Rossby Centre wind fields do not complain about that.

As our results show, and we write a few lines later, the greatest negative bias is found in the Western Baltic Sea. Also for the RCO simulations, there were negative biases in the Western Baltic Sea (Meier et al., 2004). For the rest we refer to our answer of the major comment to the same topic above.

Line 158: the mean bias of the model results improves significantly does not make sense as the bias can become smaller but cannot be improved; please reformulate.

Replaced "improves" with "reduces".

Lines 161162: please reformulate default simulations to make clear that simulations forced with unadjusted winds from atmospheric models are meant. It is strongly suggested to use unadjusted or similar instead of default that could be misleading also in what follows. The same applies to legends of Fig. 4 and to numerous occasions below.

See our previous response on this topic.

Line 169: use simply: stations (Fig. 4) shows.

Changed.

Line 179180: the ensemble is close to the observed return levels is heavy jargon and ambiguous; please explain what exactly is meant.

We deleted this sentence since we found it to be redundant.

Page 9, Caption to Fig. 3: explain STD; reformulate with and without the increased wind speed, the rest of the caption repeats information given in the text and should be removed or radically shortened.

We adjusted the caption accordingly: "Mean bias of the model's ESLs and standard deviation summarised (STD) for each station of Fig. 2 for simulations listed in Tab. 1."

Line 196: one can understand what variability between the 30-year return levels means but a classic way of saying this is differences between the estimates of 30-yr return levels.

Changed to "differences".

Page 11, lines 201203: In short, comparing the 30-year return levels for each station, the patterns are similar except for the generally lower estimates for the 30-year return levels. This sentence says nothing.

You are right. The sentence misses the reference to the GEV results. Rephrased to: "In short, comparing the 30-year return levels for each station, the patterns are similar to the GEV results except for the generally lower estimates for the 30-year return levels."

References to Fig. 6 appear before references to Fig. 5.

Corrected.

Line 207: probably pattern obtained using is meant.

Rephrased the sentecen to "The general pattern of high and low GPD 30-year return levels (Fig. 5c) is very similar to the pattern of high and low GEV 30-year return levels in Fig. 5a."

Line 208: the 95% confidence interval of the return levels from the GPD method is similar to the results  please reformulate for clarity.

Rephrased to "Also, the spatial GPD 95% confidence interval pattern and the values are similar to the GEV 95% confidence interval (Fig. 5b and d)."

Lines 211212: please reformulate for clarity.

Reformulated to: "This claim is supported by the deviations of the 30-year GPD return levels from the ensemble mean (Fig. S4 in the Supplement). Each ensemble member shows similar deviations (both in pattern and values) to the GEV return levels in Fig. **??**."

Page 12, caption to Fig. 5: remove Spatial.

Is removed.

Page 14, lines 214215: the comparison of ESLs with observations and the estimated 30-year return levels differ significantly  a comparison cannot differ; please reformulate.

Reformulated to: "We have seen in the previous sections that the modelled ESLs differ from observations to a different extents for each atmospheric forcing. Therefore, the estimated 30-year return levels differ significantly within the ensemble, leading to significant uncertainty."

Lines 224225: However, how the size and shape of the different areas of positive and negative trends differ, e.g. the extent of the area in the Western Baltic Sea that shows a positive trend seems unfinished, please reformulate for clarity.

Reformulated to: "However, the size and shape of the areas of positive and negative trends differ, e.g. the extent of the area in the Western Baltic Sea that shows a positive trend varies for each ensemble atmospheric forcing."

Lines 237238: something is missing in The variability of the atmospheric high wind speeds (i.e. storms) in the different atmospheric datasets forward into the ESLs.

We removed the sentence.

Line 259: To reduce the uncertainty, a large regional ensemble is needed to minimise the errors contains two basically independent claims.

Rephrased to "To reduce the uncertainty, we believe a large regional ensemble is needed."

Page 18, line 278: tX should denote the trend seems jargon; consider saying the value of partial derivative tX characterizes the slope of the trendline or similar.

Changed to "[...] where the value $\partial_t X$ denotes the slope of the trend line."

> Line 280: it is indeed true that directional changes and preconditioning must play a critical role; however, it might make sense to even more strongly discuss with (Soomere and Pindsoo, 2016) and (Pindsoo and Soomere, 2020) who derive a fairly rapid trend for annual maxima of preconditioning using the RCO model with a specific (gust-adjusted) wind fields. This kind of manipulation with wind data might be a good additional member of similar ensembles in the future compared to straightforward increase in the wind speed.

We added a sentence on the trends of the preconditioning: "Whereas Soomere and Pindsoo (2016) and Pindsoo and Soomere (2020) find a clear positive trend of up to $10\,\mathrm{mm}\,\mathrm{yr}^{-1}$, the ensemble of this study shows a negative trend. They further show that preconditioning is responsible for up to $4-5\,\mathrm{mm}\,\mathrm{yr}^{-1}$ of the total trend, showing that preconditioning indeed plays a crucial role."
This second part of this comment goes into the direction of the adjustment of wind speeds or how wind speeds are converted to wind stress via the drag coefficient. Thus, we refer to our response to the comments of reviewer 2 on this topic.

> Line 309: strongly modified by the default increase in wind speeds remains unclear.

Rephrased to "Nevertheless, these would be strongly modified by the adjusted wind speeds and may lead to over- or underestimation of these. This would have consequences for the general dynamics, e.g. stratification or the total salt import by Major Baltic inflows."

> Lines 320 and 321: distribution needs not to be capitalized.

changed

> Line 325; This uncertainty can be reduced by using much larger ensembles is probably true but still it is a conjecture (=hope), not a conclusion.

We reformulated to the sentence to "One approach trying to minimise this uncertainty can be much larger ensembles."

> Page 22, line 322: Andree et al. (2022) has been published as Nat. Hazards Earth Syst. Sci., 23, 18171834, 2023, https://doi.org/10.5194/nhess-23-1817-2023

Changed to the published version.

> Page 23, line 380: reference Grinsted, A.: is incomplete.

Must be a formatting thing with the bibtex 'inbook' item. Final type setting will hopefully fix it.

> DOI is missing from most of references and the references contain several smaller issues (e.g., capitalization of journals titles)

added DOI's where there exist DOI's.

**References**

Arns, A., Wahl, T., Wolff, C., Vafeidis, A. T., Haigh, I. D., Woodworth, P., Niehüser, S., Jensen, J., 2020. Non-linear interaction modulates global extreme sea levels, coastal flood exposure, and impacts. Nature communications 11 (1), 1–9.

Bamber, J. L., Oppenheimer, M., Kopp, R. E., Aspinall, W. P., Cooke, R. M., 2019. Ice sheet contributions to future sea-level rise from structured expert judgment. Proceedings of the National Academy of Sciences 116 (23), 11195–11200.

Gräwe, U., Burchard, H., 2012. Storm surges in the Western Baltic Sea: the present and a possible future. Climate Dynamics 39 (1), 165–183.

Gräwe, U., Klingbeil, K., Kelln, J., Dangendorf, S., 2019. Decomposing mean sea level rise in a semi-enclosed basin, the Baltic Sea. Journal of Climate 32 (11), 3089–3108.

Kiesel, J., Lorenz, M., König, M., Gräwe, U., Vafeidis, A. T., 2023. Regional assessment of extreme sea levels and associated coastal flooding along the german baltic sea coast. Natural Hazards and Earth System Sciences 23 (9), 2961–2985.
URL https://nhess.copernicus.org/articles/23/2961/2023/

Kudryavtseva, N., Soomere, T., Männikus, R., 2021. Non-stationary analysis of water level extremes in Latvian waters, Baltic Sea, during 1961–2018. Natural Hazards and Earth System Sciences 21 (4), 1279–1296.

Meier, H. E. M., Kniebusch, M., Dieterich, C., Gröger, M., Zorita, E., Elmgren, R., Myrberg, K., Ahola, M. P., Bartosova, A., Bonsdorff, E., Börgel, F., Capell, R., Carlén, I., Carlund, T., Carstensen, J., Christensen, O. B., Dierschke, V., Frauen, C., Frederiksen, M., Gaget, E., Galatius, A., Haapala, J. J., Halkka, A., Hugelius, G., Hünicke, B., Jaagus, J., Jüssi, M., Käyhkö, J., Kirchner, N., Kjellström, E., Kulinski, K., Lehmann, A., Lindström, G., May, W., Miller, P. A., Mohrholz, V., Müller-Karulis, B., Pavón-Jordán, D., Quante, M., Reckermann, M., Rutgersson, A., Savchuk, O. P., Stendel, M., Tuomi, L., Viitasalo, M., Weisse, R., Zhang, W., 2022. Climate change in the Baltic Sea region: a summary. Earth System Dynamics 13 (1), 457–593.
URL https://esd.copernicus.org/articles/13/457/2022/

Meier, H. M., Broman, B., Kjellström, E., 2004. Simulated sea level in past and future climates of the Baltic Sea. Climate research 27 (1), 59–75.

Pindsoo, K., Soomere, T., 2020. Basin-wide variations in trends in water level maxima in the Baltic Sea. Continental Shelf Research 193, 104029.

Soomere, T., Pindsoo, K., 2016. Spatial variability in the trends in extreme storm surges and weekly-scale high water levels in the eastern Baltic Sea. Continental Shelf Research 115, 53–64.

Suursaar, Ü., Sooäär, J., 2007. Decadal variations in mean and extreme sea level values along the Estonian coast of the Baltic Sea. Tellus A: Dynamic Meteorology and Oceanography 59 (2), 249–260.

Wahl, T., Haigh, I. D., Nicholls, R. J., Arns, A., Dangendorf, S., Hinkel, J., Slangen, A. B., 2017. Understanding extreme sea levels for broad-scale coastal impact and adaptation analysis. Nature communications 8 (1), 1–12.

Wolski, T., Wiśniewski, B., 2021. Characteristics and long-term variability of occurrences of storm surges in the Baltic Sea. Atmosphere 12 (12), 1679.

**Author response to reviewer 2 comments**

Marvin Lorenz and Ulf Gräwe

September 8, 2023

We thank both referees, Tarmo Soomere and one anonymous reviewer, for their detailed comments and thoughts. Our responses to reviewer 2 are given below.

> This paper presents extreme sea level analysis for the Baltic Sea, based on a GETM simulation of the water elevation dynamics for the years 1979-2018. The paper is well written and addresses an important topic. However, certain aspects of the research should be clarified, particularly related to the ocean model configuration, before the manuscript can be recommended for publication.

Thank you. We answer your comments in the following.

> Major comments:
> The model configuration, presented in Section 2.1 should be elaborated. The authors should provide sufficient information about the model configuration such that the simulations can be reproduced by independent researchers. In particular, as atmospheric pressure and wind stress are key drivers for extreme water levels, used forcing methods should be presented in more detail. Which wind stress formulation was used? Why that particular one was chosen? Can it be argued that the existing and widely-used formulations are suitable for the Baltic Sea (to the best of my knowledge, many wind stress formulations are designed for global ocean simulations). These details should be added and their impact also discussed in the manuscript.
> In this study, the wind speeds have been artificially increased by 3 to 11 percent to take into account the fact that atmospheric models have a tendency to underestimate extreme wind speeds. Altering the wind stress parametrization would be another way of achieving the same, potential benefit being that one can only alter the wind stress for high wind speeds.

In GETM, the wind stress is computed with:

$$\vec{\tau} = C_D \rho_{\text{air}} |\vec{u}_{10}| \vec{u}_{10},$$

with $\rho_{\text{air}} = 1.25$ kg m$^{-3}$ and the 10m wind speed vector $\vec{u}_{10}$. The drag coefficient $C_D$ is computed by the formulation of Kara et al. (2000). In our barotropic simulations we

do not consider temperature differences between the ocean and atmosphere. The drag coefficient reads as

$$C_D = 10^{-3} \left( 0.862 + 0.088w - 0.00089w^2 \right),$$

where this parameterization limits the wind speed $w = \max(2.5, \min(32.5, |\vec{u}_{10}|))$. You are right that this information is really important for the reader, thus we included this into the model description:
"The wind stress is calculated from the $10\,\mathrm{m}$ wind fields with

$$\vec{\tau} = C_D \rho_{\mathrm{air}} |\vec{u}_{10}| \vec{u}_{10}, \tag{1}$$

with $\rho_{\mathrm{air}} = 1.25$ kg m$^{-3}$ and the $10\,\mathrm{m}$ wind speed vector $\vec{u}_{10}$. The drag coefficient $C_D$ is computed by the formulation of Kara et al. (2000). In our barotropic simulations we do not consider temperature differences between the ocean and atmosphere. Therefore, the drag coefficient reads as

$$C_D = 10^{-3} \left( 0.862 + 0.088w - 0.00089w^2 \right),$$

where the wind speed is limited: $w = \max(2.5, \min(32.5, |\vec{u}_{10}|))$."
Since this formulation considers higher drag coefficients for higher wind speeds, we believe the main variability and the main reason for the necessary winds speed adjustments lie in the atmospheric datasets. Concerning your point on the validity of such formulation based on a parameterization for global ocean models for the Baltic Sea, there are parameterizations based on near shore measurements (Smith et al., 1992). However, if these different formulations perform better or worse have to be tested in the future.
We added a new paragraph to the discussion where we briefly elaborate your two raised points: "There exist many formulations, i.e. polynomial fits, of the drag coefficient which translate wind speeds into winds stresses (e.g. Kara et al., 2000; Large and Yeager, 2009). Here we have used the formulation of Kara et al. (2000). Using a different formulation, e.g. one that yields a larger drag coefficient for high wind speeds, could have improved the representation of ESLs and potentially decreased the wind factors we applied. Since there exist many formulations, each formulation would lead to different ESLs, thus introducing a source for variability. The different formulations depend on the geographic locations of the underlying data which is fitted, e.g. offshore versus onshore (Smith et al., 1992). The formulation we have used (Kara et al., 2000) is based on observations made in the Arabian Sea. One could argue that this formulation may not suitable for the Baltic Sea. However, we expect for this study that the variability of the atmospheric datasets, Figs. 9 and 10, to be much larger than the differences in ESLs due to the drag coefficient formulations. Still, the exploration of the effect of different drag coefficient formulations on ESLs in the Baltic Sea should be carried out in a future study. A calibrated drag coefficient formulation using an adjoint model could be a possible solution (Peng et al., 2013). However, depending on the wind input fields, different best fits have to be expected. Therefore, this inverse method cannot be expected to provide a one-size-fits-all solution."

In order to obtain reliable extreme value estimates, the model should be sufficiently accurate in representing the tides, emptying and filling of the Baltic Sea basin, seiche waves, and atmospherically-driven effects - extreme SSH values are formed as a superposition of all of these. (While the tides might not play a major role in the Baltic Sea itself, they may affect the volume flux to/from the North Sea.) The manuscript focuses mostly only on the atmospheric effects. It would be good to discuss the other effects and their impact as well.

Our model does include all of the above processes except tides, which are not important in the Baltic Sea (Gräwe and Burchard, 2012). The emptying and filling of the Baltic is also atmosphere driven, thus included in the results. However, the disentanglement of these processes is not trivial as they interact with each other, e.g. seiche-surge interaction can increase ESLs by up to 10cm in the Baltic Sea (Arns et al., 2020). Also, their statistical distributions are different. Thus, the GEV- and GPD approach may not work well, as mentioned by the other reviewer. Since this point goes into the direction of a point also raised by reviewer 1, we included a paragraph on these points into the discussion: "ESLs are the superposition of many sea level processes: preconditioning, storm surges, and standing waves (seiches). For example, since preconditioning can increase the mean sea level in the Baltic Sea on weekly time scales, the total sea level during a storm surge can be increased. These two (and more) compounding mechanisms follow different statistical distributions (e.g. Suursaar and Sooäär, 2007). Our analysis neglects this point as the GEV and GPD methods assume a single statistical distribution which still led to good fits. Furthermore, the mentioned mechanisms are interacting non-linearly (Arns et al., 2020). Disentangling these two (and more) compounding processes of this ensemble remains a matter for a future study."

Section 3 shows a comparison of extreme sea levels. To better assess the skill of the model, I recommend including a comprehensive statistical analysis of the reference model SSH in the Baltic Sea (and perhaps the Danish waters). It would then be clearer whether we are only testing the effects of different atmospheric forcings using a poor SSH model, or whether the model reproduces SSH dynamics well in general (giving more confidence that combined SSH effects can be reproduced).

We added the Root Mean Square Errors and correlation coefficients between the modelled and observed time series for each tide gauge to the Supplement (see Fig. 1 which is the new Figure S1) and referred to it in Section 3.1.:
"The time series comparison for each tide gauge station shows a good agreement with low Root Mean Square Errors (RMSE $\leq 0.1\,\mathrm{m}$) and high Pearson correlation coefficients $R \approx 0.9$, see Fig. S1."
And in the Supplement:
"In addition to the ESL comparison in the main text, we compare here the full length time series of the tide gauge stations with the different model runs. We compare the Root

Mean Square Error (RMSE),

$$\text{RMSE} = \sqrt{\frac{1}{N}\sum_{i=1}^{N}\left(\eta_i^{\text{obs}} - \eta_i^{\text{mod}}\right)^2}, \tag{2}$$

and the Pearson correlation coefficient $R$,

$$R = \frac{1}{N}\sum_{i=1}^{N}\frac{(\eta_i^{\text{mod}} - \bar{\eta}^{\text{mod}})(\eta_i^{\text{obs}} - \bar{\eta}^{\text{obs}})}{\sigma^{\text{mod}}\sigma^{\text{obs}}}, \tag{3}$$

where $\eta_i$ denotes the discrete time series of the observed sea level and the modelled sea level, respectively, $\bar{\eta}$ denotes the temporal mean of the respective time series, and $\sigma$ denotes the respective standard deviation. For all simulations the correlation coefficients are all around 0.9 and most RMSEs are smaller than 0.1,m (Fig. 1). For stations in the Kattegat, the $R$-values are smaller since our simulations excluded tides which are still present in this area. Also for the tide gauges in coastal lagoons, e.g. Althagen, the correlation is much smaller since the sea level dynamics cannot be captured correctly due to the coarse resolution."

> The model uses a constant bottom roughness length value (z0 = 1 mm) for all the simulations. Is this a realistic value? How was it chosen? Was the same value used of all nesting levels? Tuning the bottom friction for North Sea-Baltic Sea simulations is not a trivial task (e.g. Kärnä et al. (2023) and Kärnä et al. (2021)) if one wishes to represent SSH dynamics well across the domain - it also affects the attenuation of seiche oscillations in the Baltic.

We have used $z_0 = 5$ mm for the North Atlantic nest and indeed $z_0 = 1$ mm for the North Sea / Baltic Sea nest. You are right that a constant bottom roughness is not the most realistic. However, past studies have intensively calibrated and validated the model setup and led to the value of 1 mm, e.g. for sea level dynamics and ESLs (Gräwe and Burchard, 2012), but also for Major Baltic Inflows (MBI, e.g. Mohrholz et al., 2015) (in a baroclinic version of this setup). Correctly reproducing the timings and strengths of MBIs is very sensitive to the choice of the bottom roughness parameter. The spatially constant $z_0$ is capable to do that. Therefore, we believe the constant value is suitable for our application. Kärnä et al. (2021) also use a constant bottom roughness of 1mm for the Baltic Sea. Nevertheless, future studies, especially going to higher and higher resolution, have to revisit this topic.

> I highly encourage the authors to share the GETM source code, input files, and also post-processing scripts for reproducibility. Quite often some data (e.g. forcings) cannot be shared due to licence restrictions, but even under such circumstances, the authors should provide links to where the data can be accessed.

We included links to the code and data now in the "code and data availability"-statement.

[Figure]

Figure 1: Comparison of the Root Mean Square Error (RMSE) and the Correlation coefficient between the different model runs for each tide gauge station. Note that the values between the adjusted wind speed simulations (black) and the default wind speed simulations (blue) are very similar. Therefore the black dots are hidden behind the blue dots.

In the light of the given results, it seems that understanding the sensitivity of extreme sea level values to wind forcing is essential. Such sensitivities can be computed directly with adjoint models (e.g. Kärnä et al. (2023) and references therein). As such adjoint/inverse modeling could be an important asset in this research in addition to more traditional ensemble methods.

This seems indeed to be a good idea. We added this point to the discussion in two paragraphs: "Nevertheless, these would be strongly modified by the default increase in wind speeds and may lead to over- or underestimation of these with consequences for the general dynamics, e.g. stratification or the total salt import by Major Baltic inflows. One possible way of tackling this problem could be adjoint/inverse models (Errico, 1997) which can be used to calibrate coastal ocean input fields like bottom friction (Kärnä et al., 2023) or in this case wind speed factors."
"Still, the exploration of the effect of different drag coefficients formulations on ESLs in the Baltic Sea should be carried out in a future study. A calibrated drag coefficient formulation using an adjoint model could be a possible solution (Peng et al., 2013). However, depending on the wind input fields, different best fits have to be expected. Therefore, this inverse method cannot be expected to provide a one-size-fits-all solution."

Minor comments: Which GETM version was used to run the simulations?

The version is GETM 2.5. A frozen version is now linked in the code availability statement. Now included in the main text.
"We use the General Estuarine Transport Model (GETM (version 2.5), Burchard and Bolding, 2002), a structured coastal ocean model (Klingbeil et al., 2018), to simulate the surface elevation in the Baltic Sea."

line 98: Please elaborate. Are you only using the local atmospheric pressure information to calculate an additional offset to boundary SSH value? This would not include any atmospheric pressure-driven waves.

Only at the open boundary of the North Atlantic setup we prescribe the sea level by the inverted barometric pressure formula, i.e. the sea level is computed from the pressure above the respective water columns. You are right that we exclude waves which were generated outside the domain. We modified the sentence to: "Along its boundary, air pressure-induced water level changes are imposed using the atmospheric pressure from ERA5 to include large pressure systems from the Atlantic into the model chain (inverted barometric effect)."

line 105: The atmospheric forcing is not the same for the North Atlantic and the Baltic Sea models. How can you ensure that SSH at the nesting boundary agree, e.g. if a low pressure system is located in a different place in ERA5? This could generate spurious strong wave fronts in the model and skew the ESL analysis.

This is true but we expect the effect of this to be very small because of following reasons: 1. The Danish Straits function as a low-pass filter thus the high-frequency shock waves would be filtered. The same reason why tides are negligible in the Western Baltic Sea. 2. The boundary is far away from the Baltic Sea and the North Sea provides enough surface area for the nested setup's forcing to overwrite these errors by their wind stresses and pressure fields, respectively. 3. ERA5 is the only dataset that spatially covers the domain completely.

We mention the first two arguments now in the main text: "This simulation prescribes boundary conditions for the one nautical mile North Sea / Baltic Sea domain (all simulations). This may introduce some inconsistencies in the sea level when using other atmospheric forcings for the North Sea / Baltic Sea nesting stage. However, we assume these errors to have little effect on Baltic Sea ESLs since the Danish Straits act as a low-pass filter. Thus, storm surges are generated inside the Baltic Sea."

line 154: For the sake of clarity, please emphasize how you calculate ESLs from model/observation SSH. E.g. add an equation.

We rephrased these sentences and added additional information:
"For this comparison, we use the German Federal Maritime and Hydrographic Agency's definition of a storm surge in the Baltic Sea, which defines a storm surge as a sea level of more than 1 metre above mean sea level. Furthermore, we only consider events that are separated by more than 48 hours. We search for events that fulfil these criteria in the observed tide gauges and compare them with the modelled sea level for each station."

Figure 4: I would not use a line plot here. There's no continuity between the stations (at least in the sense the figure suggests). A bar or violin plot for each station would be better.

Changed to a bar plot.

Figure 7 caption: if you mention a) mention also b)

fixed

line 288: typo FThe

fixed

line 322: remove "therefore", there's no causality here

removed

line 330: The reviewers should have access to the data/source code during the review.

We will do in the future!

**References**

Arns, A., Wahl, T., Wolff, C., Vafeidis, A. T., Haigh, I. D., Woodworth, P., Niehüser, S., Jensen, J., 2020. Non-linear interaction modulates global extreme sea levels, coastal flood exposure, and impacts. Nature communications 11 (1), 1–9.

Burchard, H., Bolding, K., 2002. GETM: A General Estuarine Transport Model; Scientific Documentation. European Commission, Joint Research Centre, Institute for Environment and Sustainability.

Errico, R. M., 1997. What is an adjoint model? Bulletin of the American Meteorological Society 78 (11), 2577–2592.

Gräwe, U., Burchard, H., 2012. Storm surges in the Western Baltic Sea: the present and a possible future. Climate Dynamics 39 (1), 165–183.

Kara, A. B., Rochford, P. A., Hurlburt, H. E., 2000. Efficient and accurate bulk parameterizations of air–sea fluxes for use in general circulation models. Journal of Atmospheric and Oceanic Technology 17 (10), 1421–1438.

Kärnä, T., Ljungemyr, P., Falahat, S., Ringgaard, I., Axell, L., Korabel, V., Murawski, J., Maljutenko, I., Lindenthal, A., Jandt-Scheelke, S., Verjovkina, S., Lorkowski, I., Lagemaa, P., She, J., Tuomi, L., Nord, A., Huess, V., 2021. Nemo-nordic 2.0: operational marine forecast model for the baltic sea. Geoscientific Model Development 14 (9), 5731–5749.
URL https://gmd.copernicus.org/articles/14/5731/2021/

Kärnä, T., Wallwork, J. G., Kramer, S. C., 2023. Efficient optimization of a regional water elevation model with an automatically generated adjoint.

Klingbeil, K., Lemarié, F., Debreu, L., Burchard, H., 2018. The numerics of hydrostatic structured-grid coastal ocean models: state of the art and future perspectives. Ocean Modelling 125, 80–105.

Large, W., Yeager, S., 2009. The global climatology of an interannually varying air–sea flux data set. Climate dynamics 33, 341–364.

Mohrholz, V., Naumann, M., Nausch, G., Krger, S., Grwe, U., 2015. Fresh oxygen for the Baltic Sea  an exceptional saline inflow after a decade of stagnation. Journal of Marine Systems 148, 152–166.
URL https://www.sciencedirect.com/science/article/pii/S0924796315000457

Peng, S., Li, Y., Xie, L., 2013. Adjusting the wind stress drag coefficient in storm surge forecasting using an adjoint technique. Journal of Atmospheric and Oceanic Technology 30 (3), 590–608.

Smith, S. D., Anderson, R. J., Oost, W. A., Kraan, C., Maat, N., De Cosmo, J., Katsaros, K. B., Davidson, K. L., Bumke, K., Hasse, L., et al., 1992. Sea surface wind stress and drag coefficients: The HEXOS results. Boundary-layer meteorology 60, 109–142.

Suursaar, Ü., Sooäär, J., 2007. Decadal variations in mean and extreme sea level values along the Estonian coast of the Baltic Sea. Tellus A: Dynamic Meteorology and Oceanography 59 (2), 249–260.

---

## Author Response (AR2)

**Author response to reviewer comments**

Marvin Lorenz and Ulf Gräwe

October 13, 2023

We thank Ilker Fer and both referees, Tarmo Soomere and one anonymous reviewer, again for their comments and time. Our responses are given below.

**1    Response to Ilker Fer**

- Both reviewers are satisfied with your revision in the previous round and recommend minor clarifications. In addition to addressing their comments, please also include an estimate of the accuracy/uncertainty of the observational data in the relevant section.

Reponse We have added following sentences to Section: "2.2 Observational Gauge Data": "From both sources we have obtained quality controlled sea level data of hourly frequency. The sea level data is accurate to within one centimetre."

**2    Responses to Tarmo Soomere**

- The authors have taken into account almost all my suggestions, except for adjusting or more exactly specifying what is meant under "default". Thus, I am basically happy with the result and suggest the manuscript for publication, with a few very minor technical adjustments.

- Lines 52-54: it is of course true that "In general, sea-level rise shifts the ESLs (relative to present-day mean sea level) to higher base levels, and thus high ESLs become more frequent (Wahl et al., 2017)." However, it might be mentioned (for less informed readers) that projections of ESL events based, e.g., on generalized extreme value distributions contain the mean sea level as a linear term (e.g., Coles et al., 2001) and thus the frequency of ESL with respect to the new mean will not change. This remark gives inter alia an additional justification of the assumption on line 112: "Furthermore, the mean sea level is kept constant in order to study only the atmospheric-induced extreme sea levels themselves" and also for the procedure of de-meaning on lines 121-122.

Response You are right. We have slightly changed the sentence in the introduction to "In general, sea-level rise shifts the ESLs (relative to present-day mean sea level) to higher base levels, and thus high ESLs become more frequent, although surge events themselves may not have changed their frequencies (Wahl et al., 2017)."

For the sentence in the Methods section we have changed it to: "Furthermore, the mean sea level is kept constant in order to study only the atmospheric-induced extreme sea levels themselves (ESL distributions treat the mean sea level as a linear term Coles et al. (2001))."

- It seems important to me adjust one detail: It is strongly recommended to define around Line 120 (that says "default wind speeds") that default wind speed is the one that is taken directly from the atmospheric models without any adjustment; just to avoid misinterpretation.

Response This is a very good point to avoid confusion. We changed the sentence to: "For each atmospheric forcing, we ran one simulation with default wind speeds (directly taking the values provided by the atmospheric dataset) and one with increased wind speeds, see Tab. 1, for a total of 13 ensemble members."

- Line 170: should be Fig. 3.

Response Fixed.

- Lines 283-284: "These two (and more) compounding mechanisms follow different statistical distributions (e.g. Suursaar and Soor, 2007)." It is indeed true that Suursaar and Soor mentioned that the set of sea level values at Prnu "cannot be fitted by any single reasonable theoretical distribution" and that "the distribution /—/ probably consists of several distributions due to mixed populations." These distributions were extracted for the eastern Baltic proper and Gulf of Finland in [Soomere, T., Eelsalu, M., Kurkin, A., Rybin, A. 2015. Separation of the Baltic Sea water level into daily and multi-weekly components. Continental Shelf Research, 103, 2332, doi: 10.1016/j.csr.2015.04.018].

Response We have added to the sentence: "These two (and more) compounding mechanisms follow different statistical distributions (e.g. Suursaar and Sooäär, 2007) which can be extracted by separating the time series into different temporal components (Soomere et al., 2015)."

- Line 408 Nature communications should be capitalized

- Line 421: I guess it should be 208 pages

- Line 445: Grinsted, A.: is part of BACC II.

Response The bibtex entry includes BACC II as the book title, but the reference style does not show it in the pdf. I think the type setter will fix this.

- Line 475: this is from Special Issue No 85

- Line 478-479: Kärnët al. is incomplete

Response These are all the information that can be obtained from arXiv

- Line 480: Climate dynamics should be capitalized

- Line 501: Climate research should be capitalized

- Line 508: Peltier, 2004: journal title etc. is missing

- Line 531: Boundary-layer meteorology should be capitalized

- Line 539: Scientific reports should be capitalized

- Line 545: Nature communications should be capitalized

- Line 554: volume and page numbers are missing

Response Fixed the references

**3 Reponses to reviewer 2**

The authors have addressed my comments in the revised manuscript. I only have a few minor remarks before the paper can be recommended for publication.

- Line 173: For the ESL comparison, ... I would prefer to have the ESL definition already in the methods section 2 (or at least before presenting the results in Fig. 3).

Response We have moved it to Section "2.2 Observational Gauge Data" and have slightly rephrased the sentence: "For the direct comparison of ESL events with observations (Section 3.1), we use the German Federal Maritime and Hydrographic Agency's definition of a storm surge in the Baltic Sea. It defines a storm surge as a sea level of more than 1 metre above mean sea level."

- Line 362: [...] Therefore, this inverse method cannot be expected to provide a one-size-fits-all solution. Arguably the existing bulk wind stress formulae are indeed a generalisable "one-size-fits-all" solution. They have been calibrated using a realistic ocean simulation and a sufficiently large wind forcing data set. This is, in fact, a form of inverse modeling; The only difference is that the tuning is (presumably) done manually. Therefore, there is no fundamental reason why the adjoint method could not be used in the same fashion to provide generalisable wind stress formulae. It all depends on how the inverse model is configured, the extend and duration of the simulation, cost function etc. The above statement, therefore, is not quite valid and I would remove it.

Response You are right. We have deleted the sentence.

**References**

Coles, S., Bawa, J., Trenner, L., Dorazio, P., 2001. An introduction to statistical modeling of extreme values. Springer.

Soomere, T., Eelsalu, M., Kurkin, A., Rybin, A., 2015. Separation of the Baltic Sea water level into daily and multi-weekly components. Continental Shelf Research 103, 23–32. URL `https://www.sciencedirect.com/science/article/pii/S0278434315001077`

Suursaar, Ü., Sooäär, J., 2007. Decadal variations in mean and extreme sea level values along the Estonian coast of the Baltic Sea. Tellus A: Dynamic Meteorology and Oceanography 59 (2), 249–260.

Wahl, T., Haigh, I. D., Nicholls, R. J., Arns, A., Dangendorf, S., Hinkel, J., Slangen, A. B., 2017. Understanding extreme sea levels for broad-scale coastal impact and adaptation analysis. Nature Communications 8 (1), 1–12.